# Loss of *CFHR5* function reduces the risk for age-related macular degeneration

Mary Pat Reeve [1,2,3,12], Stephanie Loomis[4,12], Eija Nissilä[5], Thomas W. Soare [6], Tobias Rausch [7], Zhili Zheng [2,3], Pietro DELLA BRIOTTA PAROLO[2,3], Daniel Ben-Isvy [3,8,9], Elias Aho[5], Emilia Cesetti [5,10], Yoko Okunuki[4], Helen McLaughlin[4], Johanna Mäkelä[11], FinnGen*, Mitja Kurki[1,2,3], Michael E. Talkowski [3,8], Jan O. Korbel [7], Kip Connor[4], Seppo Meri[5], Mark J. Daly [1,2,3,13] ✉ & Heiko Runz [1,4,6,7,13] ✉

Age-related macular degeneration (AMD) is a prevalent cause of vision loss in the elderly with limited therapeutic options. A single chromosomal region around the complement factor H gene (*CFH*) is reported to explain nearly 25% of genetic AMD risk. Here, we used association testing, statistical finemapping and conditional analyses in 12,495 AMD cases and 461,686 controls to deconvolute four major *CFH* haplotypes that convey protection from AMD. We show that beyond *CFH*, two of these are explained by Finn-enriched frameshift and missense variants in the *CFH* modulator *CFHR5*. We demonstrate through a FinnGen sample recall study that *CFHR5* variant carriers exhibit dose-dependent reductions in serum levels of the *CFHR5* gene product FHR-5 and two functionally related proteins at the locus. Genetic reduction in FHR-5 correlates with higher complement activation capacity and a thicker retinal photoreceptor layer. Our results propose therapeutic downregulation of FHR-5 as promising to prevent or treat AMD.

Age-related macular degeneration (AMD) is characterized by a slowly progressive loss of central vision due to the death of retinal pigment epithelium and photoreceptor cells in the retina. The prevalence of AMD increases with age, and in Western countries it affects about one in five individuals over the age of 90[1]. While efficient treatments exist for the neovascular form of AMD (wet AMD), the medical need remains substantial for the much more common dry AMD, which distinguishes itself by macular atrophy and a progressive buildup of subretinal deposits known as drusen.

Twin and genome-wide association studies (GWAS) have established that up to 71% of AMD susceptibility is heritable[2,3], with the most recent meta-analyses identifying over 60 genome-wide significant loci[4,5]. Out of these, a single chromosomal region around the complement factor H gene (*CFH*) has been reported to explain up to 25% of the genetic risk for dry and wet AMD[3]. This association, together with other loci including *C3, C5, C9, CFI*, and *CFD*, has consolidated a central role of the complement system in AMD pathogenesis. By now, multiple lines of evidence propose that the overactivation of the alternative

[1]Institute for Molecular Medicine Finland (FIMM), University of Helsinki, Helsinki, Finland. [2]Analytic and Translational Genetics Unit, Department of Medicine, Massachusetts General Hospital, Boston, MA, USA. [3]Program in Medical and Population Genetics, Broad Institute of Harvard and MIT, Cambridge, MA, USA. [4]Research and Development, Biogen Inc, Cambridge, MA, USA. [5]Department of Bacteriology and Immunology, Translational Immunology Research Program, University of Helsinki, Helsinki, Finland. [6]insitro Inc., South San Francisco, CA, USA. [7]European Molecular Biological Laboratories (EMBL), Heidelberg, Germany. [8]Center for Genomic Medicine, Massachusetts General Hospital, Boston, MA, USA. [9]Division of Medical Sciences, Harvard Medical School, Boston, MA, USA. [10]Department of Biomedical Sciences, Humanitas University, Milan, Italy. [11]Finnish Biobank Cooperative (FinBB), Turku, Finland. [12]These authors contributed equally: Mary Pat Reeve, Stephanie Loomis. [13]These authors jointly supervised this work: Mark J. Daly, Heiko Runz. *A list of authors and their affiliations appears at the end of the paper. ✉e-mail: mjdaly@broadinstitute.org; heiko.runz@gmail.com

pathway (AP) C3 convertase, on which the classical, alternative and MBL-lectin complement pathways converge, initiates a proinflammatory cascade that results in retinal tissue damage. Activation of C3 can be inhibited by Factor H (FH), the protein product of *CFH*, which slows down complement activation. Elevating FH has been suggested as a strategy for the treatment of AMD[6]; however, this approach is challenged by the need to keep regional FH levels within a narrow margin for an optimal efficacy and safety profile.

*CFH* was the first disease susceptibility locus ever unveiled through the GWAS approach[7–10]. Since its discovery, the complexity of this locus has increased with increasing GWAS sample sizes and geographic representation. Conclusive evidence of two very strong, independent variants at *CFH*[11,12] was published shortly after discovery, including the original association to the common *CFH* missense variant p.Tyr402His, which in functional studies was shown to affect binding of FH to its interaction partners and increase AMD risk[13,14]. Similarly, rare loss-of-function variants in *CFH,* such as p.Arg1210Cys were later demonstrated to accelerate the onset of AMD by several years[15]. However, *CFH* variants modulate susceptibility to AMD in conjunction with different haplotypes that include *CFH* together with a set of five adjacent and highly homologous genes termed *CFH*-related genes 1–5 (*CFHR1-5*)[7,16,17]. The products of *CFHR1-5*, FHR1-5, are believed to compete with FH for C3b and other target binding and counteract its attenuating effect on complement activation[18,19]. Consistently, structural variants (SVs) resulting in deletions of *CFHR1* and *CFHR3*[16] or of *CFHR1* and *CFHR4*[20], respectively, have been proposed to reduce AMD risk, although the independence of such effects from *CFH* has remained contested[21,22]. Recently, through a coding variant meta-analysis in over 650,000 individuals, we identified a frameshift variant within *CFHR5* (p.Glu163insAA, rs565457964) as associated with a strong protection from AMD[23], although also for this signal, no independence from other variation at the locus has yet been established.

Here, we explored the relative contribution of *CFHR5* to the risk for AMD and its subforms through refined association testing and fine-mapping analyses for AMD in FinnGen (FG), a large population biobank cohort in Finland[24]. We show that carrying a haplotype that leads to genetic loss of *CFHR5* function is associated with a reduced risk for AMD independent of other evident association signals at the *CFH* locus. We further demonstrate that *CFHR5* frameshift variant carriers not only exhibit reduced blood levels of FHR-5, but also of FHR-2 and FHR-4 proteins. Moreover, carriers have a higher capacity to activate the classical and alternative complement pathways, consistent with enhanced abilities of the complement system to clear retinal debris, as well as preserved photoreceptor layers in their retinas. Our results establish downregulation of *CFHR5* as an attractive opportunity for targeted AMD therapies.

## Results

### FinnGen GWAS reveals 23 regions independently associated with AMD

We conducted an association study for AMD (https://risteys.finregistry.fi/endpoints/H7_AMD, including both wet and dry AMD) in FinnGen (FG) data freeze (DF) 12 (12,495 cases, 461,686 controls) across the full allele-frequency spectrum. This identified genome-wide significant associations to SNPs at 23 independent genomic regions (Supplementary Fig. 1; Supplementary Data 1). 21 of these loci fall into regions that had been reported previously as associated with AMD or its subforms[3,5], while two appear to be novel (rs759283, rs74026308). Of the 62 loci identified as associated with AMD in a recent multi-ancestry meta-analysis[5], 19 loci met genome-wide significance criteria in FG, 31 replicated at nominal significance ($p < 0.05$) and 11 did not replicate (with one locus not being included in the FG analysis set) (Supplementary Data 2). Systematic finemapping and conditional analyses prioritized likely causal variants within three out of five FG

GWAS regions with multiple apparently independent signals. This included a single base pair insertion in an intron of *LIPC* (15:58430391:G:GC; $p_{cond} = 7.92 \times 10^{-8}$; minor allele frequency [MAF] = 0.23) in an established AMD region on chromosome 15. Our analysis confirmed variants at the *HTRA1/ARMS2* locus (chr10:120954932-123954932) as the genome-wide strongest drivers of genetic association with AMD. Notably, however, while a previously described[25]. independently associated missense variant (p.Ala69Ser; 10:122454932:G:T; $p = 3.07 \times 10^{-490}$) continues to provide credibility for *ARMS2* as a causal gene at this locus, we also identified a strong, conditionally independent association with a rare, Finnish-enriched (~40-fold relative to non-Finnish Europeans) insertion in the coding region of *HTRA1* (10:122461686:C:TCCT; $p_{cond} = 1.34 \times 10^{-18}$). This variant leads to the introduction of a single serine residue within a string of leucine residues (see Methods) and can be assumed to modulate HTRA1 protein functions, which possibly contributes to increasing AMD risk.

Consistent with previous AMD GWAS, we identified strong conditionally independent associations near several genes that encode members of the complement pathway, including *CFH*, *C3* and *CFI*. Out of these, *CFH* showed the second strongest association with AMD in Finns, with nearly 100 regional SNPs exhibiting $p$ values below $10^{-375}$. Multiple studies have robustly established two independent variants in *CFH* as key contributors to this signal, one being tagged by rs1410996 and the other by the coding missense variant p.Tyr402His[11,12]. Consistently, proxies for both variants also showed the strongest signals in our analysis, with $p$ values of $9.6 \times 10^{-618}$ (for rs10922109; "FG$_{AMD}$1") and $2.0 \times 10^{-590}$ (for rs570618; "FG$_{AMD}$2"), respectively, in univariate analysis (Fig. 1; Supplementary Data 3). We also replicated our earlier[23] finding of a strong association between the low-frequency frameshift variant p.Glu163insAA in *CFHR5* (chr1:196994128:C:CAA, rs565457964, *CFHR5*$_{fs}$) with protection from AMD ($p = 1.1 \times 10^{-68}$ in univariate analysis) at this locus. Notably, a phenome-wide scan in carriers of this frameshift variant, which is ~10-fold enriched in Finns relative to UK Biobank (UKB), in the latest FG release revealed significant associations only with protection from AMD and its subforms dry and wet AMD, but none of the 2404 other phenotypes ascertained in FG DF12 (Supplementary Fig. 2, Supplementary Data 4). The strong GWAS signal in the absence of evident signs for adverse events in FG motivated us to further de-risk *CFHR5* through refined genetic and functional studies for its potential as a drug target to prevent or treat AMD.

### Finemapping CFH region reveals conditionally independent protective CFHR5 variants

In order to determine whether *CFHR5*$_{fs}$ or any other regional variants were independently associated with AMD risk or protection, we performed conditional analyses in FG using REGENIE on a 1.15 Mb region bounded by recombination hotspots (chr1:196,200,000–197,350,000; hg38) and spanning 12 genes (*KCNT2, CFH, CFHR3, CFHR1, CFHR4, CFHR2, CFHR5, F13B, ASPM, SEPT14P12, ZBTB41* and *CRB1*). For consistency with an earlier AMD GWAS, we fixed the same top two variants identified in Fritsche et al.[3], rs10922109 and rs570618, which are in complete LD ($r^2 > 0.99$) with the originally reported variants rs1410996 and p.Tyr402His, respectively, to begin the conditional search for independently associated variants. Unsurprisingly, both SNPs explained the majority of the AMD signal in the region, also in our analysis. However, conditioning on both variants left a number of tertiary variants with conditional p-values between $5.9 \times 10^{-7}$ and $6.2 \times 10^{-25}$ (Fig. 1; Supplementary Data 3) within a smaller ~380 kb region encompassing *CFH* and all five *CFHR* genes. Within this region we found a third independent signal (p_cond=$6.2 \times 10^{-25}$, beta = -0.50, "FG$_{AMD}$3") tagged by rs537634973, a roughly 2% allele frequency (AF) SNP enriched in Finns and in high LD with five other similarly associated variants, including missense variants in *CFHR5* (rs139017763, p.Gly278Ser, $r^2 = 0.91$) and *CFHR2* (rs79351096, p.Cys72Tyr, $r^2 = 0.85$). Notably, *CFHR5* p.Gly278Ser is also in high LD ($r^2 = 0.72$) with the

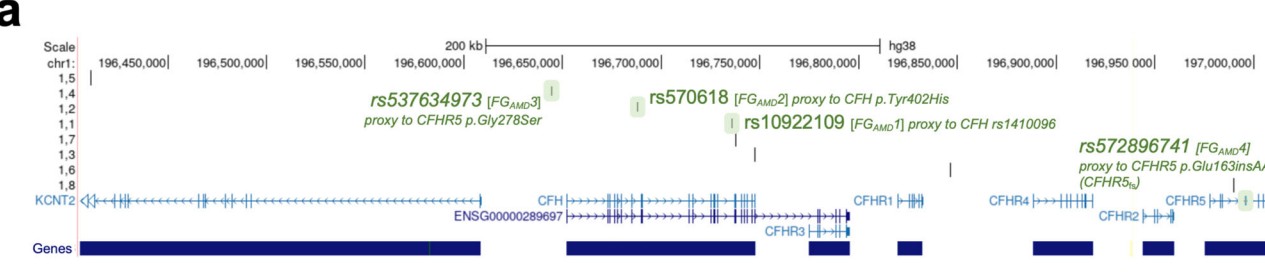

**Fig. 1 | Conditional analysis of the *CFH* region reveals four major independent AMD signals.** Finemapping and conditional analyses of genome-wide significant lead signals (FG lead) within the *CFH* region (chr1:196,328,000–197,014,000; hg38) from a FinnGen (DF12) genome-wide association study (GWAS) for age-related macular degeneration (AMD) with 12,495 cases and 461,686 controls (see Supplementary Data 1 and https://r12.finngen.fi/pheno/H7_AMD). **a** Positions of coding genes, FG GWAS lead SNPs, and previously analyzed[3,17] regional AMD GWAS signals (1.1–1.8). Conditionally independent signals (FG_AMD1–4) in high linkage disequilibrium with presumably causal *CFH* and *CFHR5* variants are highlighted in green. **b** Association *p* values from univariate and two rounds of conditional (cond) analyses. "True" associations representing the major conditionally independent signals are highlighted in green. For details, see Methods and Supplementary Data 3. "GWAS lead" is based on Fritsche et al.[3], MAF minor allele frequency, n.a. not analyzed, n.i. not imputed.

fourth-strongest signal reported as independently driving the regional AMD association in Fritsche et al.[3].

We repeated the conditional analysis, including this third signal. This led to another genome-wide significant signal ($p_{cond}$ = 6.8 × 10⁻¹¹, beta = -0.29, "FG_AMD4") tagged by rs572896741 and in high LD with five other regional variants of ~4% AF and enriched up to 10-fold in FG relative to UKB. Notably, this fourth signal included CFHR5_fs ($r^2$ = 0.91) that, like *CFHR5* p.Gly278Ser, was associated with protection from AMD, indicating a potential allelic series in *CFHR5* that would further raise its attractiveness as a target for AMD therapies.

No other signals within the 1.15 Mb *CFH* region reached genome-wide significance after conditioning on all four signals, although we confirm two additional signals reported earlier[3,17] as conditionally independent beyond these four signals with $p < 1 × 10⁻⁶$ (Supplementary Data 3).

## No evidence for an impact of regional SVs on AMD risk in Finns

The *CFH* gene family resides within a genetically unstable region that has evolved from several incomplete segmental duplication events and harbors SVs of variable population frequency[20]. Several of these SVs have been reported to reduce or increase the copy numbers of one or more *CFHR* genes[8,20]. To ensure that SVs of potential relevance to the association evidence were adequately considered in our conditional analyses, we leveraged an SV map compiled from whole genome sequencing (WGS) data in more than 64,000 individuals from gnomAD (v4 release) and identified SNP proxies for two well-established *CFH* region deletion variants, a common deletion of 91 kb encompassing *CFHR1* and *CFHR3*, and a low-frequency, Finn-enriched 121 kb deletion of *CFHR1* and *CFHR4*. Importantly, neither of these two SVs was in LD

with the credible sets used in finemapping, nor did they show a residual signal after our stepwise conditional analysis. For instance, conditioning on the *CFH* lead signal rs1410996 reduced the strength of association between the FG lead variant rs61818925 (signal 1.6 in Fig. 1b), a tight proxy for the *CFHR1/CFHR3* deletion ($r^2$ = 0.98), and AMD from 6.0 × 10⁻¹⁶⁵ to $p$ = 1.3 × 10⁻¹⁰, and further to $p$ = 0.29 upon jointly conditioning on rs1410996 and *CFH* p.Tyr402His. Our conditional analyses thus strongly propose that the often-studied large deletions in the *CFH* region do not substantively influence AMD risk in Finns. Because these are common deletion variants, this provides little support for the hypothesis that variants impacting *CFHR1*, *CFHR3* and *CFHR4* coding regions contribute substantially to the strong association between the *CFH* locus and AMD, while conversely, our results support an eminent role of high-impact variants in *CFH* and *CFHR5* genes.

## No evidence for cryptic regional variation to explain CFHR5_fs effects

To further validate whether *CFHR5* variants are indeed independently protective for AMD or potentially act in conjunction with yet concealed novel variation at the locus, we next leveraged phased short-read and long-read sequencing data from the expanded 1000 Genomes (1KG) project[26,27] and studied the *CFH* regional architecture at base-level resolution across ethnicities. We identified 37 individuals carrying a total of 38 *CFHR5* coding variants among the 6,404 chromosomes 1 haplotypes in this cohort (Supplementary Data 5). Among these, eight carried the p.Glu163insAA (CFHR5_fs) variant in a heterozygous state and one in a homozygous state. Six were heterozygous carriers of another frameshift variant at the identical position,

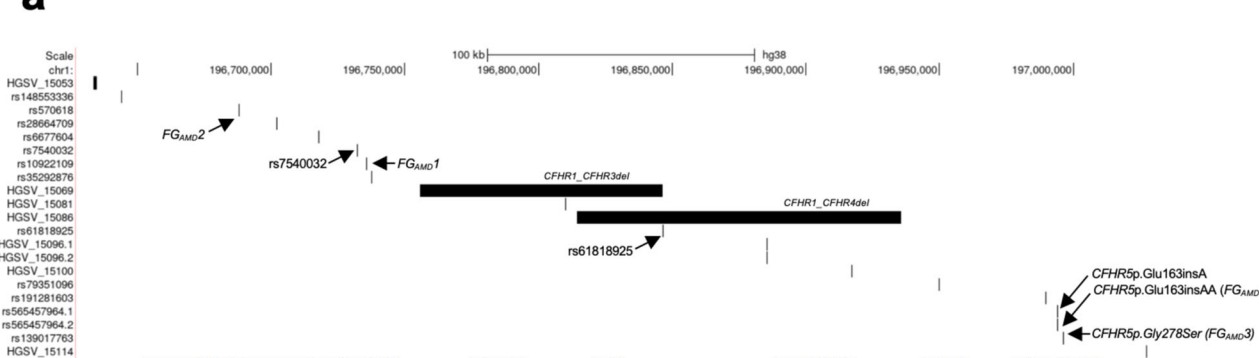

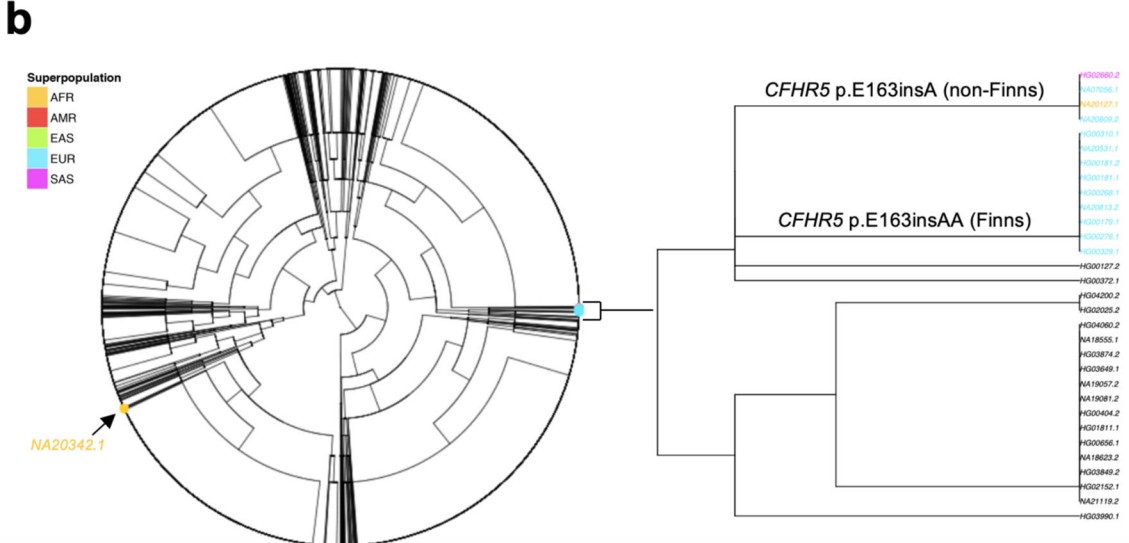

**Fig. 2 | Phased short- and long-read sequencing excludes structural variants as a driver of *CFHR5* AMD associations. a** Positions of coding genes, lead AMD GWAS SNPs (from this study and Lores-Motta et al.[17]) and structural variants (SVs) with allele frequencies >1% identified through short- and long-read sequencing of the ~380 kb *CFH* region (chr1:196,640,000–197,020,000; hg38) in 6404 chromosomes of the 1000 Genomes cohort (1KG). Previously described large deletions encompassing *CFHR1* and *CFHR3* (HGSV_15069), as well as *CFHR2* and *CFHR4*

(HGSV_15086), respectively, are highlighted together with their proxy SNPs and identified coding variants in *CFHR5*. **b** Dendrogram of *CFH* region haplotypes in 1KG. Carriers of *CFHR5* frameshift variants and their ethnicity (color) are highlighted. Haplotypes 1 and 2 are indicated by sample suffixes, i.e., NA20342.1 corresponds to haplotype 1 of sample NA20342. For details, see Method and Supplementary Data 5–10.

p.Glu163insA, and 22 carried the missense variant p.Gly278Ser. We further identified 12 carriers of a rare *CFHR5* intronic variant, rs191281603-G, that was proposed earlier as associated with AMD[17], but that did not meet independent significance in our conditional analyses. Consistent with our previous findings[23], *CFHR5* p.Glu163insAA was substantially more prevalent in unrelated Finns (7/198; AF = 3.54%) relative to all other unrelated 1KG participants (2/1670; AF = 0.12%) (Supplementary Data 6), with the only non-Finnish carriers being two Tuscans from Italy and one African American individual. Also, p.Gly278Ser is enriched in Finns (6/198; AF = 3.03%) relative to the rest of the unrelated 1KG cohort (4/1670; AF = 0.24%), while only one of the rs191281603-G and none of the p.Glu163insA carriers were Finnish.

Interrogation of the *CFH* region (chr1:196,640,000–197,020,000; hg38) with phased WGS data across all available samples identified 10 SVs with AF > 1% (Fig. 2a; Supplementary Fig. 3; Supplementary Data 7). This included three common SVs: the above-studied 91 kb *CFHR1*/*CFHR3* deletion (HGSV_15069; AF = 27.2%) as well as two insertions of 97 bp (HGSV_15081; AF = 20.7%) and 280 bp (HGSV_15100; AF = 20%), respectively. Rarer SVs in the region stretched from the

above-studied *CFHR1*/*CFHR4* deletion (HGSV_15086; AF = 1.9%) to a duplication of 64 bp (HGSV_15114; AF = 1.8%). None of these SVs impacted coding regions of *CFH* or *CFHR1-5* genes.

Taking into account our FG lead signals, SVs with AF > 1% and eight lead SNPs proposed as independent from published AMD GWAS[3,17], we identified 254 different haplotypes in the *CFH* region across the full 1KG cohort (Fig. 2b; Supplementary Data 8–10). This breadth of haplotypes could be traced to just three foundational haplotypes that in Finns showed frequencies of 15%, 40% and 38%, respectively. Phased long-read data confirmed that the remaining 7% haplotypes are derivatives of the most common Finnish haplotype and contained either *CFHR5*fs or *CFHR5* p.Gly278Ser. Notably, despite being found exclusively in non-Finnish individuals, the haplotype structure of p.Glu163insA carriers was identical to that of *CFHR5*fs and *CFHR5* p.Gly278Ser carriers, suggesting that similar mutational events must have led to their occurrence. Nevertheless, other evolutionary mechanisms may also exist for *CFHR5* coding variants to arise, as evidenced by the single 1KG participant of African ancestry (NA20342) who carried *CFHR5*fs on an alternative haplotype.

**Table 1 | Recall study cohort demographics by participating biobank**

| | Biobank | | | | Total (N = 399) |
|---|---|---|---|---|---|
| | Auria (N = 81) | Borealis (N = 109) | Eastern Finland (N = 101) | Tampere (N = 108) | |
| *N* AMD cases | 39 | 60 | 48 | 52 | 199 |
| Mean age at sampling (SD) [years] | 70.4 (10.0) | 76.6 (7.6) | 74.0 (7.1) | 75.5 (8.3) | 74.4 (8.5) |
| Min; max | 47; 91 | 61; 91 | 62; 92 | 56; 96 | 47; 96 |
| Mean age at AMD diagnosis (SD) [years] | 75.6 (10.5) | 74.3 (8.3) | 72.4 (7.1) | 72.4 (10.2) | 73.6 (9.1) |
| Min; max | 58; 92 | 58; 90 | 61; 88 | 55; 94 | 55; 94 |
| Percent female [%] | 31 | 61 | 44 | 63 | 51 |
| *CFHR5*fs variant status | | | | | |
| C/C (non-carrier, n/c) | 62 | 74 | 73 | 73 | 282 (70.7%) |
| C/CAA (het erozygote) | 19 | 28 | 18 | 22 | 87 (21.8%) |
| CAA/CAA (homozygote) | 10 | 7 | 0 | 13 | 30 (7.5%) |
| AMD x *CFHR5*fs variant status | | | | | |
| AMD, *CHFR5*fs n/c | 37 | 49 | 48 | 48 | 182 (45.6%) |
| AMD, *CFHR5*fs het | 2 | 11 | 0 | 4 | 17 (4.3%) |
| AMD, *CFHR5*fs hom | 0 | 0 | 0 | 0 | 0 (0%) |
| no AMD, *CHFR5*fs n/c | 25 | 25 | 25 | 25 | 100 (25.1%) |
| no AMD, *CFHR5*fs het | 17 | 17 | 18 | 18 | 70 (17.5%) |
| no AMD, *CFHR5*fs hom | 10 | 7 | 0 | 13 | 30 (7.5%) |

*SD* standard deviation, *n/c* non-carrier, *het* heterozygote, *hom* homozygote.

Importantly, haplotype-resolved long-read sequencing in 1KG identified no additional cryptic rare or common variation that would be expected to modify protein-coding elements within the *CFH* region. Taken together, our base-level interrogation of the *CFH* region finds no reason to suggest that the association with reduced risk for AMD in *CFHR5* frameshift and missense variant carriers is conveyed by local SVs or concealed single base pair changes other than those within *CFHR5* or its associated haplotypes.

## CFHR5 frameshift variant carriers show lower FHR-5, FHR-2 and FHR-4 blood levels

To test for a relevance of $CFHR5_{fs}$ on FHR-5 levels and function, we established a process to recall samples from broadly consented FG participants in a customized manner (Supplementary Fig. 4). In partnership with the Finnish Biobank Cooperative (FinBB), we selected serum samples from 200 individuals with a diagnosis of dry AMD based on registry information that had been archived at four Finnish biobanks. We matched these cases to 200 controls based on age, sex, *CFH* and *CFHR5* variant status and the absence of alternative diagnoses affecting the eye (Table 1 and Methods). The final recall study cohort had an average age of $73.6 \pm 9.1$ and $74.4 \pm 8.5$ years at diagnosis and blood sampling, respectively. Thirty participants (7.5%) were homozygous for the $CFHR5_{fs}$ variant, 87 (21.8%) were heterozygous, and 282 (71%) were non-carriers (Supplementary Data 11, 12). No AMD cases homozygous for $CFHR5_{fs}$ were available for resampling

We measured the serum levels of 6627 circulating proteins, including FH and FHR1-5, using the SomaScan platform[28]. One sample failed quality control, leaving a final sample size of 399. Serum levels of FHR-5 were reduced in homozygous and in heterozygous $CFHR5_{fs}$ carriers in a dose-dependent manner ($p = 4.72 \times 10^{-71}$ after controlling for age at sampling, sex and AMD status), with no residual levels detectable in homozygotes (Fig. 3a). There was no evidence for batch effects by biobank (Supplementary Fig. 5). Findings were confirmed with three different FHR-5 somamers (Supplementary Fig. 6), as well as independently through Western Blot (Supplementary Fig. 7) and validate that p.Glu163insAA indeed abolishes FHR-5 serum protein levels. Notably, also levels of FHR-2 and FHR-4 were found to be reduced

dose-dependently in $CFHR5_{fs}$ carriers ($p = 9.85 \times 10^{-24}$ and $p = 1.23 \times 10^{-7}$, respectively, after controlling for age at sampling, sex and AMD status; Fig. 3a; Supplementary Data 13). Conversely, no differences were found for FH and FHR-3 ($p > 0.05$), while FHR-1 showed a nominally significant ($p = 0.03$) increase in the recall setting.

We next tested whether the effects of $CFHR5_{fs}$ status on serum protein levels synergized with other variation at the *CFH* locus. For this, we grouped the 500,348 genotyped FG participants (DF12) according to haplotypes based on the four regional AMD signals that had remained significant after our stepwise conditional analyses. Five risk groups were defined based on whether individuals carried only risk alleles (0-0-0-0) or were heterozygous (=1) or homozygous (=2) for one or several of the four protective variants in the *CFH* region. As expected, individuals who carried risk alleles for all four signals showed the highest probability to be diagnosed with AMD (OR = 1.67), which is reflected by 56.2% of AMD cases in FG falling into this highest risk category (Fig. 4). Conversely, individuals who carry either *CFHR5* p.Gly278Ser or $CFHR5_{fs}$ on top of protective alleles at *CFH* rs1410996 and p.Tyr402His show the highest protection from AMD ($p\_cond = 1.5 \times 10^{-18}$, OR = 0.59 and $p\_cond = 2.7 \times 10^{-9}$, beta = 0.52, respectively). Notably, among the recall study participants, the dose-dependent reduction of FHR-5 ($7.55 \times 10^{-19}$), FHR-2 ($p = 1.33 \times 10^{-5}$) and FHR-4 ($2.63 \times 10^{-3}$), but not of FH, FHR-1 and FHR-3 ($p > 0.05$) remained significant in individuals who carried one (2-2-0-1; $n = 34$) or two (2-2-0-2; $n = 30$) copies of $CFHR5_{fs}$ relative to zero (2-2-0-0; $n = 26$) copies on the most protective haplotype (Fig. 3b, Supplementary Data 14). This demonstrates that $CFHR5_{fs}$ conveys additive effects on serum protein levels that go beyond the effects from other protective variations on its respective *CFH* haplotype. A significant reduction in FHR-2, FHR-4 and FHR-5 could also be seen in carriers of *CFHR5* p.Gly278Ser (Supplementary Data 15).

To confirm the robustness and generalizability of the proteomic findings in our recall study, we sought to validate the effect of $CFHR5_{fs}$ on reducing FHR-2, FHR-4 and FHR-5 blood levels also in a wider FG cohort not enriched for AMD from which serum samples had been analyzed both, with the Somalogic panel ($N = 881$) as well as an independent proteomics platform (Olink; $N = 1732$). This yielded highly

**a**

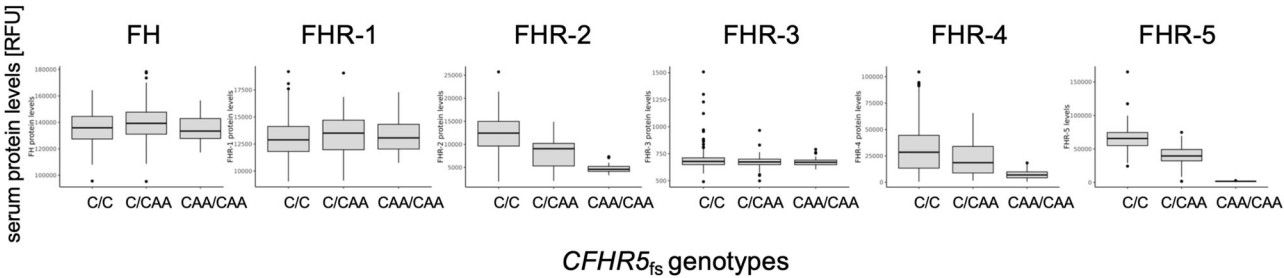

**b**

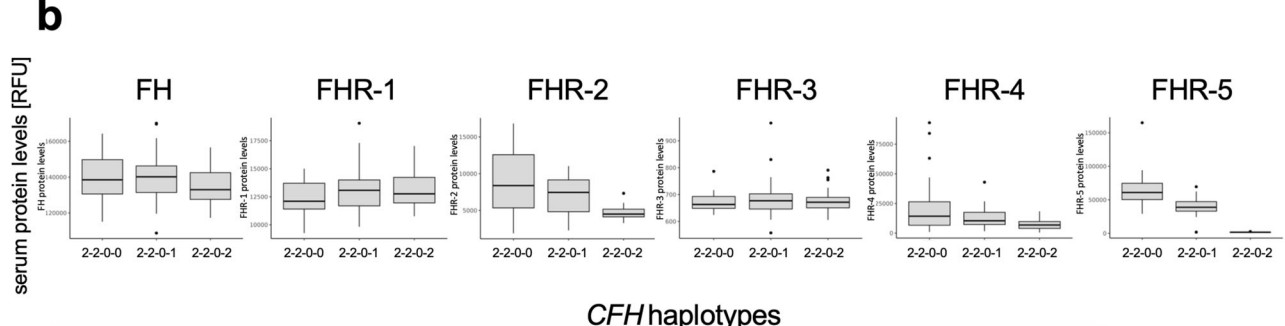

**Fig. 3 | *CFHR5*fs is linked to lower FHR-5, FHR-2 and FHR-4 blood levels in a dose-dependent manner.** Relative protein levels [in relative fluorescence units, RFU] of complement factor H (FH) and complement factor H-related factors FHR1-5 as measured by representative somamers on the SomaScan platform from serum samples of 399 FinnGen participants recalled from four Finnish biobanks (see Supplementary Data 11, 13 and 14). **a** *x* axis reflects carrier status for *CFHR5* frameshift variant p.Glu163insAA. C/C, non-carriers (*n* = 282); C/CAA, heterozygotes (*n* = 87); CAA/CAA, homozygotes (*n* = 30). **b** *x* axis reflects *CFH* haplotype status. Recall study participants with the highest genetic protection from AMD due to homozygosity for protective alleles at *CFH* rs1410996 and p.Tyr402His (2-2-0-0; *n* = 26) were compared to participants who on top carry either one (2-2-0-1; *n* = 34) or two (2-2-0-2; *n* = 30) copies of *CFHR5*fs alleles. Linear regression analyses were performed. In the box plots, the center line represents the median, the whiskers indicate standard deviations, and the dots indicate highest and lowest values.

similar results to our findings for FHR-5, FHR-2 and FHR-4 in the recall setting (Supplementary Data 16), although the initially nominally significant effect of *CFHR5*fs on modulating FHR-1 also did not replicate in the wider FG cohort. Our results are further consistent with a burden of coding variants in *CFHR5* lowering FHR-5, FHR-2 and FHR-4 plasma levels in UK Biobank (Supplementary Data 17)[29], thus replicating that genetic loss-of-function of *CFHR5* reduces FHR-5, FHR-2 and FHR-4 protein levels in blood also in an independent European cohort.

### CFHR5 frameshift carriers show increased capacity to activate complement pathways

During normal aging and early stages of AMD, complement activity is critical for the removal of retinal debris. Conversely, if overactive or misdirected, the complement system is a central driver of AMD pathology[6]. FH and FHRs are believed to be critical regulators of alternative pathway (AP) C3 convertase activity, which amplifies complement activation that was initiated via the classical (CP) and MBL-lectin pathways (LP) and converges the complement activation signal on a joint terminal pathway (Fig. 5a)[30]. To assess whether genetic loss of *CFHR5* function modulates complement activity, we measured functional activities of all three complement pathways in sera of a subset of 40 AMD patients and 44 controls from our FG recall study cohort. Notably, we found that activation capacities of both CP and AP, but not LP were significantly increased in homozygous *CFHR5*fs carriers relative to non-carriers (*p* < 0.001; Fig. 5b; Supplementary Data 18). Heterozygotes significantly differentiated from non-carriers for both CP (*p* = 0.0071) and AP (*p* = 0.0007) only in AMD cases, but not in unaffected controls (Supplementary Fig. 8), and in the overall cohort showed a tendency towards intermediate effects between homozygotes and non-carriers (Fig. 5b). As seen with protein levels, CP and

AP activation capacities relative to non-carriers (2-2-0-0; *n* = 8) remained increased in individuals who carried two (2-2-0-2; *n* = 12) copies of *CFHR5*fs when controlled for the presence of the independently protective *CFH* alleles in the region (*p* < 0.001; Fig. 5c; Supplementary Fig. 9). In summary, results from these functional analyses demonstrate that the *CFHR5*fs haplotype is independently associated with an individual's capacity to activate the complement system and confer protection from AMD.

### Retinas of *CFHR5*fs carriers show AMD-protective photoreceptor morphologies.

Thickness of the photoreceptor layer in the retina is strongly associated with the risk for AMD[31,32]. To test, whether *CFHR5*fs carriers showed morphological differences in their retinal photoreceptor layers, we conducted association testing for the four conditionally independent *CFH/CFHR5* variants with retinal parameters obtained through optical coherence tomography (OCT) imaging in UKB[33]. Consistent with previous findings[32], we found that AMD case status was associated with increased thickness of the inner segment (ELM-ISOS) and decreased thickness of the outer segment (ISOS-RPE) of the central and inner subregions of the retinal photoreceptor layer (Fig. 6, Supplementary Data 19). In carriers of AMD protective haplotypes impacting *CFH* (rs1410996 and *CFH* p.Tyr402His protective allele carriers, i.e., 1-1-0-0, 1-2-0-0, 2-1-0-0, or 2-2-0-0), these effects showed an opposite effect, as reflected by a *decreased* inner segment thickness and an *increased* outer segment thickness (Supplementary Data 20). The effects were even further pronounced in heterozygote carriers of the protective *CFHR5*fs haplotype present in UKB, no matter the concomitant *CFH* haplotype status (Supplementary Data 21). Effect sizes were the largest in *CFHR5*fs variant carriers with the highest baseline genetic protection from AMD due to homozygosity for protective *CFH*

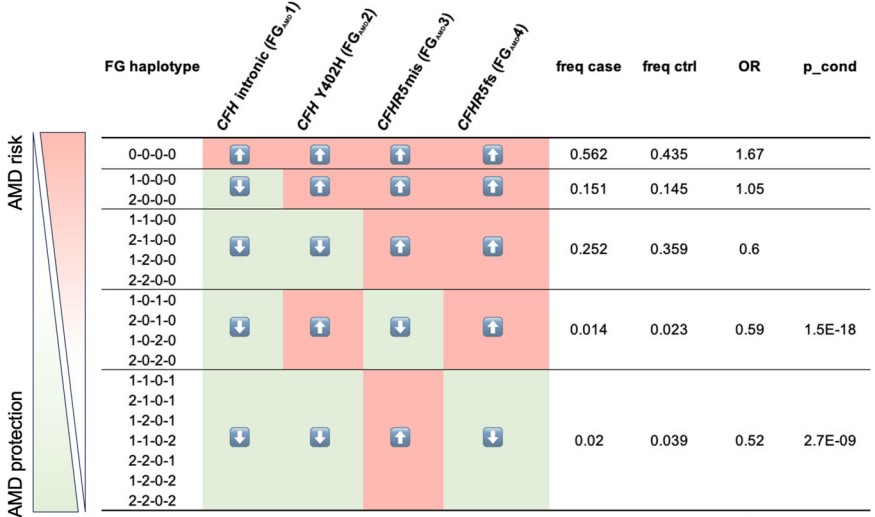

**Fig. 4 | FinnGen *CFH* regional haplotypes and relation to AMD risk in the population.** 500,348 FinnGen (FG) participants with available genotypes (DF12; see https://www.finngen.fi/en/node/2635) were grouped according to *CFH* regional haplotypes based on the four regional AMD signals that had remained significant after stepwise conditional analyses ($FG_{AMD}1-4$). Five risk groups were defined based on whether individuals carried only risk (light red/upward arrows) alleles (0-0-0-0) or were heterozygous (=1) or homozygous (=2) for one or several of the four protective (light green/downward arrows) variants in the *CFH* region. OR, odds ratio for association with a diagnostic entry for AMD (H7_AMD) based on FG health records. freq, frequency of diagnostic entry per group. p_cond, *p* value after conditioning for top two GWAS lead signals rs1410996 and p.Tyr402His in *CFH* (see Methods).

alleles at rs1410996 and p.Tyr402 (2-2-0-0; Supplementary Data 22), further supporting additive AMD-protective effects of *CFH* and *CFHR5*.

## Discussion

Here, we conducted a GWAS in nearly 475,000 Finns that replicated fifty and added two novel signals to the 62 genomic loci recently reported in a large-scale multi-ancestry GWAS for AMD[5]. Using statistical fine-mapping and conditional analyses, we undertook a fine-grained interrogation of the well-established *CFH* locus. This identified four major *CFH* haplotypes that convey protection from AMD, out of which two are linked to coding variants in the *CFHR5* gene. We demonstrate through a sample recall study that individuals carrying Finn-enriched *CFHR5* loss-of-function variants show a dose-dependent reduction in the *CFHR5* gene product, FHR-5, as well as increased activation capacity of the classical and alternative complement pathways. Carriers of *CFHR5* loss-of-function haplotypes further show AMD protective changes in the photoreceptor layer of their retinas, consistent with longer preserved vision. Our findings establish the relevance of *CFHR5* for AMD risk relative to functionally related genes at the locus and propose therapeutic downregulation of FHR-5 as a promising strategy for the prevention or treatment of AMD.

Our results elucidate the complexity of the first disease locus ever identified through the GWAS approach, which at the time required an association study with only 96 AMD patients and 50 controls[9]. Since then, *CFH* has been consistently replicated as an AMD locus, and it has become clear that the *CFH* region is characterized by substantial heterogeneity across the human population. Notably, it includes several genes with annotated functions in a disease-relevant pathway, large common SVs disrupting some of these genes, and a variable degree of risk between carriers of different regional haplotypes[3,17]. Our study showcases how such complexity can be systematically reduced to pinpoint the most relevant causal contributors to genetic disease risk within a region. For this, we leveraged statistical fine-mapping in a homogeneous population, stepwise conditional analysis of apparently independent highly significant GWAS lead signals, the exclusion of concealed regional variation through sequencing, and customized functional analyses from matched patient and control samples enriched for protective allele carriers in a recall-by-genotype setting. That

such a systematic approach to GWAS locus deconvolution is now possible is largely due to the advent of large genetically profiled biobank cohorts such as FG where low-frequency variants enriched in the population enable new discoveries, the complexity of haplotype structure is reduced due to population bottlenecks, and participants are consented a priori for dedicated follow-up analyses[23,24].

Consistent with our GWAS findings, multiple lines of evidence have established critical roles for a dysregulated complement system in the etiology of AMD for which an accumulation of lipoproteinaceous drusen in the vicinity of retinal pigment epithelium cells is considered the initial trigger[14]. In 20% of patients, this buildup of drusen is followed by a neovascular response, which can be efficiently treated with VEGF inhibitors. In contrast, patients with dry AMD show a slowly progressive loss of retinal pigment epithelium cells through cell death, leading to geographic atrophy and, through subsequent deterioration of photoreceptor cells, irreversible blindness. While in healthy individuals and early stages of the disease, an active complement system is critical for efficient removal of toxic cellular debris, its activation products are also among the most intense mediators of inflammation and can further flare up disease processes. To keep retinal cells healthy, complement activity must therefore be regulated within tight borders. FH enhances the breakdown of C3 convertase and promotes the cleavage of C3b, thereby fine-tuning the terminal complement pathway before inflammatory byproducts can be formed. The AMD risk allele p.Tyr402His is believed to reduce interaction between FH and C-reactive protein or polyanionic surfaces, leading to a misdirected activation of the terminal complement pathway[13]. Apart from genetics, the central role for FH during AMD pathogenesis has been established through biochemical, cellular, preclinical and clinical studies[34]. FHR proteins, which circulate in blood as dimers and oligomers, have been reported as having antagonistic roles to FH[16,35,36]. For instance, complete deletion of *CFHR1* and *CFHR3* through a common SV had initially been thought to be protective of AMD[16]. However, later studies showed that this deletion shares the same haplotype with the *CFH* intronic variant rs1410996 and its proxies, which are more likely to convey AMD protection[21]. Further studies have reported a direct link between elevated levels of FHR-1 and FHR-5 and a higher risk for AMD and progression to later stages of disease[36,37], with Mendelian

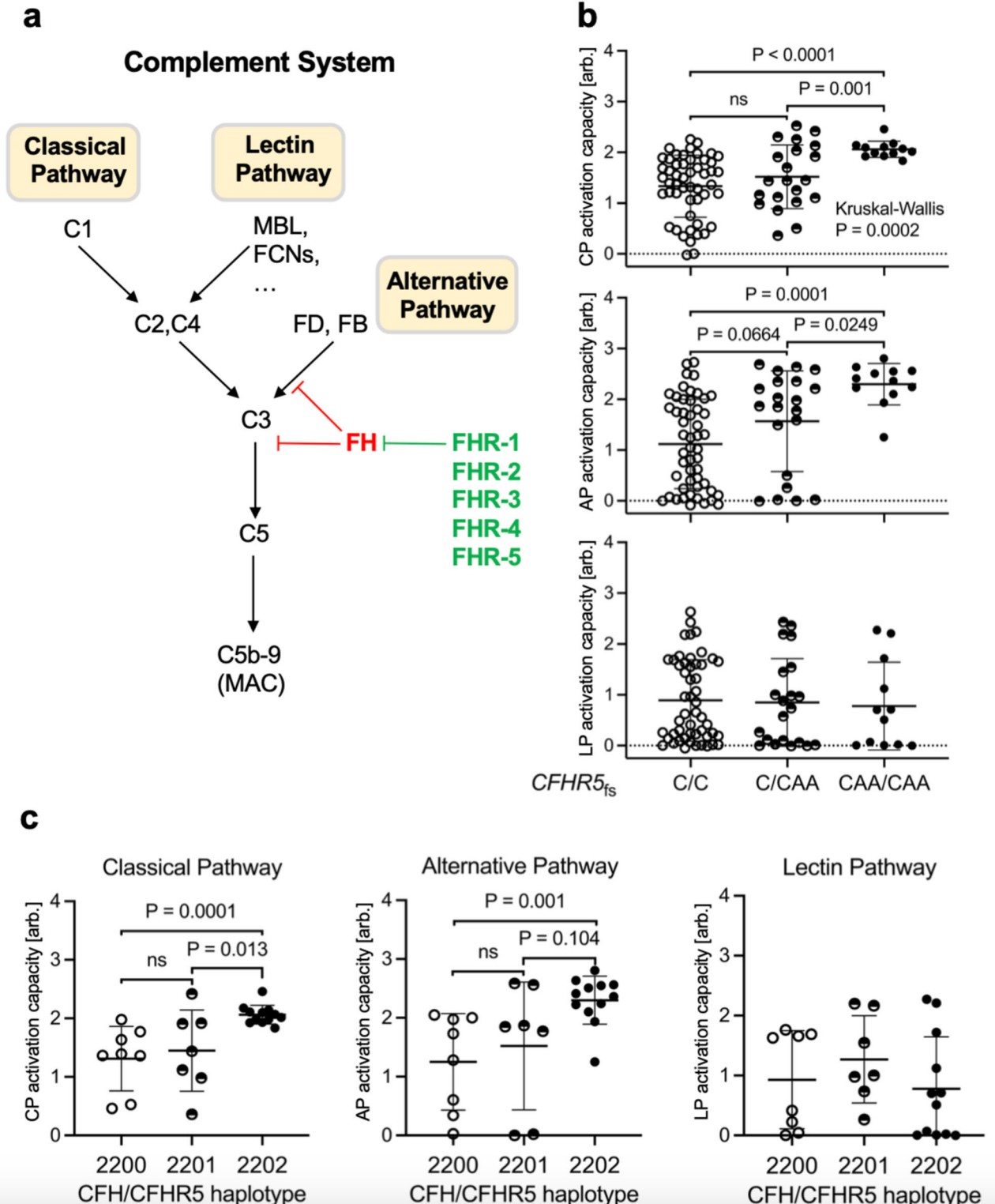

**Fig. 5 | CFHR5$_{fs}$ is linked to increased activation capacities of the classical and alternative complement pathways. a** High-level sketch of the terminal pathway of the complement system on which the classical pathway (CP), MBL-lectin pathway (LP) and alternative pathway (AP) converge. **b, c** Functional analysis of CP, AP, and LP activation capacities in serum samples from 84 recall study participants (see Supplementary Data 18). **b** *x* axis reflects carrier status for *CFHR5* frameshift variant p.Glu163insAA. C/C, non-carriers (*n* = 51); C/CAA, heterozygotes (*n* = 21); CAA/CAA, homozygotes (*n* = 12). **c** (*x* axis) reflects *CFH* haplotype status for recall study participants. Individuals with an AMD protective haplotype (2-2-0-0; *n* = 8) were compared to individuals who also carried one (2-2-0-1; *n* = 7) or two (2-2-0-2; *n* = 12) copies of *CFHR5*$_{fs}$ on this haplotype. Statistical analyses were performed with a two-sided Mann−Whitney *U* test. The Kruskal−Wallis test was applied for comparisons across all three groups. Each dot represents one individual. For each group, the center line represents the median, and the whiskers represent the standard deviation. arb. arbitrary units, Ns not significant. For details, see Supplementary Data 18.

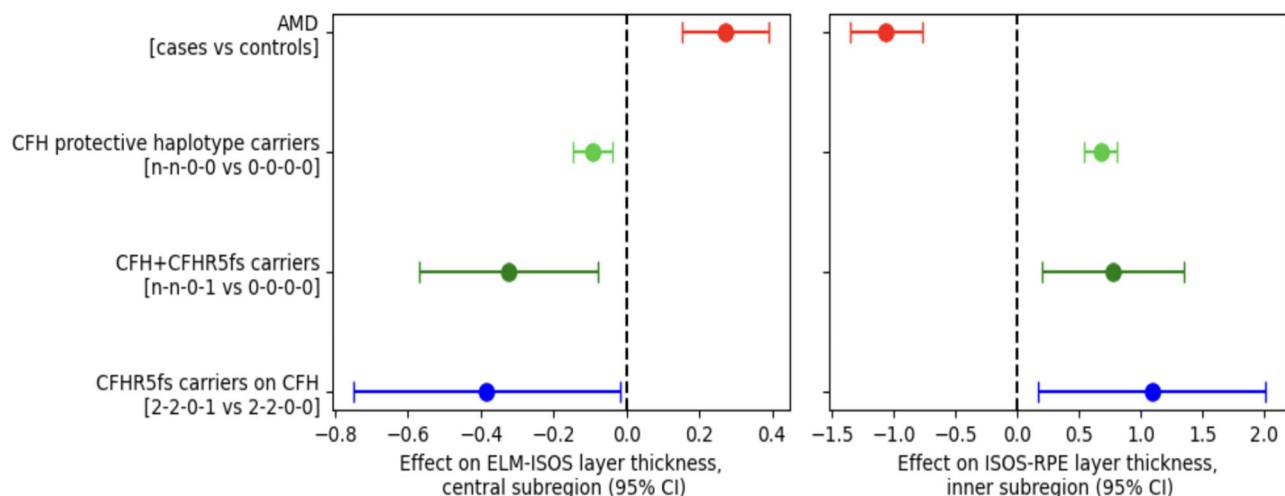

**Fig. 6 | *CFHR5*<sub>fs</sub> is associated with thickness of retinal photoreceptor layers.** Shown are median effect sizes (dots) ± 95% CI (whiskers) of associations between indicated haplotype groups with thickness of the inner (ELM-ISOS, left) and outer (ISOS-RPE, right) segments of the retinal photoreceptor layer measured from optical coherence tomography (OCT) scans in UK Biobank (UKB) participants (see also Zekavat et al.[32]). AMD cases ($n = 1083$ out of 56,359 individuals with OCT data meeting quality control criteria and data available for analysis) were defined by the ICD-10 code H35.3 (see Supplementary Data 19). *CFH* protective haplotype carriers ($n = 25,364$) were defined as UKB participants carrying at least one of the protective alleles at rs1410996 and *CFH* p.Tyr402His (n-n-0-0) (see Supplementary Data 20). *CFHR5*<sub>fs</sub> carriers ($n = 265$) are designated as either n-n-0-1 or 2-2-0-1 (see Supplementary Data 21). No individuals in the analyzed UKB cohort carried *CFHR5*<sub>fs</sub> in a homozygous state.

randomization analyses demonstrating a link between cis-regulatory GWAS variants, elevated FHR-5 levels and higher risk for advanced AMD[36,37]. This was further supported by Lorés-Motta et al. [17], who showed that certain low-frequency genetic variants in *CFHR5* that are associated with reduced FHR-5 levels are also associated with protection from AMD. Together with our own earlier findings that carriers of the rare *CFHR5* p.Glu163insAA (*CFHR5*<sub>fs</sub>) variant show a strong protection from AMD[23], these earlier studies already indicated that elevated levels of FHR-5 are associated with increased risk for AMD while reduced levels of FHR-5 are associated with a lower risk for AMD, hinting at *CFHR5* as a potentially causal contributor to AMD risk at the locus. However, none of these earlier works thoroughly assessed the effects driven by *CFHR5* variants in relation to the multifold other independent variations at the *CFH* locus that challenge fine-mapping and colocalization attempts. Strong evidence exists that several of these regional variants modulate FHR protein levels in trans, as evidenced, e.g., by reduced FHR-4 levels in carriers of the protective *CFH* rs1410996_A allele[36] and also our own findings in FG of *CFH* rs1410996_A and p.402Tyr modulating serum levels of all FHRs except FHR-3. Differences between studies assessing FHR serum levels are likely explained not only by differing sample sizes and AMD case/control status, but also by variation in haplotype composition of the respective study cohort. In our study, we thus carefully dissected the *CFH* locus through multiple rounds of fine-mapping and conditional analyses in Finns, who, as a bottlenecked population, show a reduced complexity of the regional haplotype structure and are thus particularly advantageous for deconvoluting association signals in complex GWAS regions[23]. In the case of the *CFH* region, Finns are enriched for *CFHR5* coding variants *CFHR5*<sub>fs</sub> and *CFHR5* p.Gly278Ser, with this enrichment providing a power for association testing that would necessitate multiple-fold larger sample sizes in other populations[24]. Our results firmly establish that carriers of haplotypes with *CFHR5* loss-of-function alleles show a strong, dose-dependent protection from AMD that is additive to that conveyed by common protective alleles in *CFH*, particularly rs1410996_A and p.402Tyr, which explain the majority of the association signal at the *CFH* locus[3,5]. We further demonstrate through dedicated follow-up analyses that carriers of the *CFHR5*<sub>fs</sub> haplotype show reduced FHR-5 protein levels and stronger activation capacities of the classical and alternative complement

pathways. The pronounced effects we observe cannot be explained by any other cryptic variation of protein-coding sequences at the *CFH* locus, making *CFHR5* highly probable as an independent modulator of genetic risk and a promising drug target for AMD. Our findings further shed light on how local additive effects, in this case, the co-occurrence of protective alleles in both *CFH* and *CFHR5*, may reduce genetic disease risks even further and augment association signals.

Notably, we demonstrate that *CFHR5*<sub>fs</sub> carrier status reduces serum levels of the FHR-5 protein, which in homozygous *CFHR5* "knockouts" was undetectable in our assays. The absence of FHR-5 translated into about twofold increased capacity to activate the classical and alternative complement pathways relative to non-carriers. Based on these findings, it is tempting to speculate that a higher preserved complement activity may enable carriers of variants lowering FHR-5 levels or function to more efficiently clear cellular debris in their retinas, which consequently would translate into a reduced probability of developing AMD. This is consistent with AMD-protective changes in photoreceptor layer thickness from OCT images in *CFHR5* loss-of-function variant carriers, as recently reported also by others[38]. However, our results also indicate that a mere antagonistic effect might be too simplistic a model to explain the biological interplay between FH and FHR-5. For instance, our recall study from Finnish samples cannot explain whether loss of FHR-5 function would also raise complement activity in a setting where FH is unimpaired, since all carriers of *CFHR5*<sub>fs</sub> and *CFHR5* p.Gly278Ser in the FG cohort also carried at least one of the two AMD protective alleles in *CFH*. With the emergence of both sets of variants on the same haplotype, evolution may have found a mechanism to re-adjust the balance between FH and FHR-5 in a most favorable way to optimize the complement gene family for its primary function in pathogen recognition and host immune defense[20], which could secondarily predispose to the emergence of late-onset diseases such as AMD. Alternatively, while FHR-5 has been found to inhibit binding of FH to C3b at lower concentrations than other FHRs[19], it may not act in isolation, and lower FHR-5 levels could destabilize FHR hetero-oligomers, which themselves impact FH and complement activity. In our study, *CFHR5*<sub>fs</sub> carriers not only showed reduced FHR-5, but also reduced FHR-2 and FHR-4 levels, with these effects being supported by both FG and UKB, two different proteomics platforms utilized, and for FHR-2 also extended to *CFHR5* p.Gly278Ser carriers.

Notably, our fine-grained sequence analysis of the *CFH* region did not provide evidence for other genetic variation that might explain this observation, so FHR proteins most likely regulate their mutual abundance at a post-genetic level. Further experiments in future studies are needed to clarify the exact mechanisms by which a reduction in FHR-5 increases complement activity and lowers AMD risk.

Nevertheless, already now our analyses identify *CFHR5* as an "allelic series" gene[39] that does not have evident associations with on-target safety signals that could be red flags for the development of FHR-5 directed therapies[40]. Treatments against several components of the complement pathway are being explored in clinical trials for their suitability to address, especially, the dry form of AMD. In 2023, the complement inhibitors pegcetacoplan[41] and avacincaptad pegol[42] were approved for the treatment of geographic atrophy secondary to dry AMD via intravitreal injections in the US, but not in Europe. A Phase 1 study showed that monthly injections of a recombinant full-length human FH protein (GEM103) into the eye are well tolerated[6], with preliminary results from a Phase 2 study (NCT04643886) reporting sustained C3a lowering effects. No therapies have yet targeted *CFHR*1-5 or their products. The option to reduce rather than increase its products to achieve therapeutic benefit makes *CFHR5* a potentially attractive target for neutralizing antibodies or oligonucleotides that, since FHR-5 is synthesized almost exclusively in the liver, could possibly even be explored for systemic administration. That FHR-5 lowering is accompanied by a reduction in FHR-2 and FHR-4 indicates that the requirements for a truly isoform-specific modality might not need to be so high. However, like other complement-targeting therapies, FHR-5 inhibitors might be challenged by a fairly narrow therapeutic range and a need to substantially suppress protein levels. This is supported by our finding that in *CFHR5*fs heterozygotes, preserved complement activation capacity was increased only in cases, but not controls, which suggests that partial reduction in target tissues might not suffice to fully counteract disease processes. Moreover, future studies in longitudinal genetically profiled cohorts will be necessary to better assess whether FHR-5 inhibition might indeed be suited for treatment rather than primarily prevention of AMD.

Our study has several limitations. First, while we powered the recall study sufficiently to demonstrate the effect of *CFHR5*fs on FHR protein levels and complement activation, its size was insufficient for a dissection of how less frequent and even higher resolved *CFH* haplotypes in our cohort impact these processes, which will require larger studies. Also, with its enrichment for AMD cases with available biobank samples, our recall study may not be representative of the broader population, although it is reassuring that we could replicate a reduction of FHR proteins in *CFHR5*fs carriers in the subset of the FG population with proteomics data available, as well as in UK Biobank. Notably, the distribution of *CFH* haplotypes in the recall study cohort had a more pronounced effect than AMD case-control status, which may be an important learning for the design of future such studies that for participant selection might want to take into account more granular phenotypes than the widely defined AMD code we relied on in our study. Future studies in non-European cohorts with alternative haplotype compositions may also help to further disambiguate protective effects from *CFH* versus *CFHR5*. Our discovery of a single 1KG participant of African ancestry who carries *CFHR5*fs on a different haplotype, as well as of the non-Finnish *CFHR5* frameshift variant p.Glu163insA, indicates that such informative individuals exist. A further limitation is our study design, which considered only a small fraction of the health data that an AMD patient accrues during their course of disease. More in-depth analyses in individual-level Finnish health records over an extended period of time, as well as obtaining additional samples and clinical data, such as more retinal scans from informative *CFHR5*fs carriers than we could rely on for our study from UKB participants, could help to prioritize the optimal target population for testing *CFHR5*-directed therapies.

In summary, our study exemplifies how existing and newly acquired data from a broadly consented human population cohort like FG can be leveraged and combined to inform genetic and post-genetic research well beyond the discovery of novel genetic leads. With the wealth of information captured, future research should enable a more granular understanding on how links between distinct haplotypes at GWAS loci are influenced by concomitant genetic variation elsewhere in the genome (such as the *ARMS2/HTRA1* locus), environmental contributors (such a smoking, which further increases a genetically elevated risk for AMD in *CFH* p.Tyr402His carriers[43]), or aging. It should further facilitate a more customized design of research cohorts for clinical trials, for instance, to enrich a trial population for individuals with the most informative genetic profiles to demonstrate efficacy. With permissive legislation for secondary use of research data, we predict that resources where genetic and deep longitudinal health information are linked at scale will become invaluable tools not only to better understand biology, but also accelerate the path towards efficient novel medicines.

## Methods

### FinnGen samples and participants

FinnGen (FG) (https://www.finngen.fi/en) is a public-private partnership project that aims to generate medically and therapeutically relevant insights into human disease by combining genome and digital health data from over 500,000 Finns. FG participants provide informed consent for biobank research (see below), which includes secondary use of research samples stored at nine participating Finnish biobanks. Individual-level genotypes, register data, and an overview of available samples from FG participants can be accessed by approved researchers via the Fingenious portal (https://site.fingenious.fi/en/) hosted by the Finnish Biobank Cooperative FinBB (https://finbb.fi/en/). All analyses in this manuscript rely on FG data freeze (DF) 12, for which summary-level data have been publicly released in November 2024 (https://r12.finngen.fi).

### Genotype data quality control, association testing and conditional analyses

Array-based genotype data in FG were called and subjected to variant and sample-level quality control (QC) followed by phasing and imputation as described in Kurki et al.[24]. In FG DF12, this project-wide process resulted in a total of 500,348 individuals after removal of related individuals and non-Finnish ancestry participants. This dataset was used in all FG-wide analyses in this study. GWAS analysis was conducted for phenotype H7_AMD (https://risteys.finregistry.fi/endpoints/H7_AMD), which includes individuals with at least one registry entry of either wet or dry AMD or both, comprising 12,495 cases and 461,686 controls that all passed genotype QC and inclusion criteria. GWAS analysis was performed with REGENIE 2.2.4[44] using a logistic mixed model adjusted for age, sex, genotyping batch and the first ten principal components of ancestry with an approximate Firth test for robust effect size estimation. Plink and vcf files for imputed genotypes, as well as ocular variables from the biobanks were securely transferred and stored on the DNAnexus platform. Stepwise conditional analysis of the *CFH* locus was performed identically with the serial addition of established significant variants at the *CFH* locus by adding these variants as covariates. Since associations at the *CFH* locus are highly significant for both wet and dry AMD, we utilized the combined group with any AMD diagnosis (H7_AMD) as the phenotype also for conditional analyses. Statistical fine-mapping of non-*CFH* regions was conducted using SuSIE[45]. Phenome-wide association analyses for *CFHR5*fs (p.Glu163insAA; chr1:196994128:C:CAA, rs565457964) variant carriers were conducted against a total 2405 FG DF12 phenotypes accessible via the PheWeb browser (https://r12.finngen.fi) and displayed in a LAVAA plot (https://geneviz.aalto.fi/LAVAA/) (Supplementary Fig. 2) as described in Fauman et al. [46].

Two previously reported SVs within the *CFH* region were called from Finnish participants in gnomAD version 4 (https://gnomad.broadinstitute.org) from an SV map compiled from WGS data. Proxy SNPs were identified to impute these SVs into the broader FG population, and association testing for H7_AMD was conducted as described above. We note that the conditionally independent rare, Finnish-enriched insertion variant we identified in the coding region of *HTRA1* (10:122461686:C:TCCT; $p_{cond} = 1.34 \times 10^{-18}$) has been mis-annotated in public repositories as two apparent *HTRA1* frameshift variants, 10:122461685:G:GT and 10:122461686:C:CCT. However, a look-up in primary sequencing data in gnomAD from Finns and East Asians revealed that these variants are derived from a single mutational event that replaces GC with GTCCT, thereby introducing the triplet code for a serine residue into the *HTRA1* coding sequence (10:122461686:C:TCCT).

### CFH region genetic architecture in 1000 genomes cohort

We used phased short-read and long-read sequencing data from the expanded 1000 Genomes (1KG) project[26,27] as well as assembled haplotypes from the Human Pangenome Reference Consortium (HPRC) (https://humanpangenome.org)[47] to catalog haplotype-resolved genetic variation across the *CFH* region (Supplementary Data 5–7). For the HPRC haplotypes, we used AGC[48] to extract the haplotype sequences from the AGC archive of HPRC year1 assemblies, which contains fully phased diploid assemblies for 47 samples. We then mapped the GRCh38 *CFH* region sequence to each assembly using minimap2[49] with the option -x asm10, extracted the mapped subsequence using samtools[50] and then aligned all HPRC *CFH* regional haplotypes to GRCh38. We visualized all alignments using dot plots[51] and IGV[52] to reveal divergent genetic architectures present in the haplotypes (Supplementary Fig. 3).

To call *CFH* haplotypes, we extracted eight GWAS lead SNPs from the 1KG long-read data that had been utilized previously to construct haplotypes conveying different degrees of AMD risk[17]. We further refined these haplotypes by also considering 16 additional variants that now included regional SVs with allele frequency >1% across the 1KG cohort and the GWAS lead signals from our FG DF12 AMD GWAS (Supplementary Data 7). This identified 254 different haplotypes in the *CFH* region across the 6406 chromosomes of all subpopulations. Out of these, only three showed a frequency of over 15% across all populations, 12 of >1% and 42 of >0.1%. 214 of the 254 haplotypes were present in <10 copies, with nearly 50% being singletons (Supplementary Data 8).

To estimate *CFH* haplotype frequencies across ancestries, we leveraged phased short-read data from the 1KG cohort[26] available at the International Genome Sample Resource (IGSR). We first subset the phased variants to the *CFH* region and the 2504 unrelated samples (5,008 haplotypes) and then aggregated shared haplotypes by self-identified ancestries from 26 geographic locations in five continental regions (labeled as populations and super-populations for consistency with prior 1KG literature). The 198 chromosomes from unrelated Finns in the 1KG cohort were distributed across 34 haplotypes, out of which 93% were explained by the three most common haplotypes overall, and the remainder were derivatives of these. The allele count of each *CFH* regional haplotype per ancestry is shown in Supplementary Data 9.

We also queried new, long-read 1KG data available for a subset of 1019 samples[27] to identify potentially missed SVs associated with the *CFHR5* frameshift variants p.Glu163insAA and p.Glu163insA, since short-reads have limitations in identifying SVs in repeat-rich and complex regions of the genome. Out of the 1,019 samples, one sample was a heterozygous carrier for p.Glu163insA (NA20127) and four samples were heterozygous carriers for p.Glu163insAA (HG00268, NA20342, NA20531, NA20813). For all 5 samples, we did not identify additional SVs with the long-read data within the *CFH* region, except for an intronic 86 bp deletion in four homozygous and one heterozygous carrier that resides within a tandem repeat region and shows an allele frequency of over 80% across the cohort. In summary, based on long-read data for 1,019 samples, we found no evidence for cryptic regional variation that could explain the *CFHR5*$_{fs}$ effects.

### Recall study cohort selection

Based on FinRegistry data from health records of ~5 million Finns (see: https://risteys.finngen.fi/endpoints/H7_AMD), the unadjusted period prevalence of broadly defined AMD is 1.64% in the overall Finnish population and 2.48% among the (on average older) FinnGen population. Consistent with epidemiological information from other countries, this prevalence rises steeply with older age, and among 90-year-olds in Finland, 15.1% of women and 10.0% of men have received a registry-based diagnosis of AMD. The median age at diagnosis in Finns is 79.03 years and 76.26 years in FinnGen. Thus, the prevalence of broadly-defined AMD in Finland does not seem to differ substantially from the prevalence of progressed AMD in other European populations with comparable age distributions, although we recognize that direct comparisons across national cohorts can be biased[53].

For most of the 500,348 genotyped FG participants, samples are stored at and can be retrieved from nine regional biobanks across Finland. We identified the FG subset with serum samples available in four biobanks, Auria, Borealis, Eastern Finland, and Tampere Biobank. Of these, we excluded samples with low-quality genotypes at key sites (imputation info score <0.95) and individuals with highly unlikely *CFH* haplotypes suggestive of potential genotyping errors. Of the remainder, 200 AMD cases and 200 controls were selected for a sample recall study based on individual-level phenotype data deposited at the respective biobanks. FG participants were defined as cases if they had at least one registry entry with ICD codes for dry AMD (H35.30, 2625 A) at the time of the study, as well as none of the following: ICD codes for wet AMD, hereditary conditions affecting the eye (e.g., Stargardt disease, hereditary retinal dystrophy), macular hole, macular pucker, diabetic retinopathy, glaucoma, vitreous hemorrhage, separation of retinal layers, onset of blindness at <30 years of age, or injections into the eye). Controls were required to have no ICD entries for AMD, be 65 years of age or older at the time of sampling and were matched to cases based on age and sex. To maximize our ability to examine the effect of the *CFHR5*$_{fs}$ haplotype on serum protein levels and complement activity, we originally aimed at an approximately equal distribution of *CFHR5*$_{fs}$ carriers and non-carriers between both groups. However, given an allele frequency of this frameshift variant of ~4% in the overall Finnish population together with its strong AMD protective effect, we found the allele substantially underrepresented in AMD cases. Thus, the final recall study cohort passing quality control (n = 399) included only 17 heterozygous cases (as opposed to 70 heterozygous controls) while no samples from a very small number of homozygous FG participants with diagnostic codes for dry AMD could be retrieved (as opposed to 30 homozygous controls) (Supplementary Data 11, 12).

### Proteomics data acquisition

200 μl of serum from each recall study participant was thawed, aliquoted into tubes of 130 μl for proteomic profiling and shipped on dry ice from Finland to the Somalogic lab in Boulder, CO, USA All samples underwent the Somalogic SomaScan assay to measure expression of 7000 proteins, including FH and FHR1-5. Somalogic generated a sample quality report, which flagged three of the 400 samples. Of these, only one sample was far outside of the normal QC range and hence excluded from analysis, leaving a final sample size of 399. The majority of values for FH and FHR1-5 were within the 95% normal serum RFU range provided by Somalogic, with the exception of an extreme outlier for FHR-3, which we excluded. FHR-1 and FHR-5 were both targeted by multiple somamers. One of the FHR-1 somamers (15468-14) showed evidence of binding to FH with similar affinity; thus, we used

the other FHR-1 somamer (5982-50) in our analyses. All three FHR-5 somamers showed similar QC metrics and results, so for simplicity we focused on a the somamer with the lowest serum %CV (3666-17) in our main analyses, with all three FHR-5 somamers showing similar effects on FHR-5, FHR-4 and FHR-2 levels (Supplementary Fig. 6). Statistical analyses from proteomics data were performed on the DNAnexus platform using R as described previously[28,54]. Linear regression analyses were used to determine the association between genotypes and protein levels, and between AMD and protein levels. Multi-variant analyses in recall study participants testing associations between each of the four *CFH* regional haplotypes and FH/FHR protein levels were controlled for age, sex, AMD status and the respective alleles under study.

Analyses on how carrier status for AMD-protective alleles in *CFH* and *CHFR5* impacted levels of FH and FHR1-5 in plasma of 881 FG participants profiled with the Somalogic panel, and in 1732 FG participants profiled with the Olink panel, were conducted in the FinnGen Sandbox. Results are provided as Supplementary Data 16. To identify plasma proteins associated with the burden of protein-coding variants in *CFH* and *CFHR*1-5 genes, we queried results from Dhindsa et al.[29] who had linked whole-exome sequencing data to protein levels obtained with the Olink panel from plasma of 49,736 individuals of European ethnicity as part of the UK Biobank Pharma Proteomics Project. Results for the *CFH* and *CFHR*1-5 genes from that study are provided as Supplementary Data 17.

### Complement activation capacity measurements

70 μl of serum from each recall study participant was shipped on dry ice for complement function profiling to the Finnish Institute for Molecular Medicine (FIMM) in Helsinki, Finland. Functional activation capacities of the classical, alternative and MBL-lectin complement pathways were measured using a commercial enzyme immunoassay (WIESLAB® Complement System Screen, Cat# COMPL300). This assay allows one to determine activation capacities of all three pathways by measuring the ability to generate the terminal complement complex component C5b-9, which is produced as a result of complement activation. The amount of C5b-9 generated is proportional to the potential in a sample to activate the respective complement pathways. Statistical analyses of differences between $CHFR5_{fs}$ non-carriers, heterozygotes, and homozygotes across the entire subcohort profiled for complement activity ($n = 84$) or adjusted for AMD status, as well as recall study participants with different degrees of protection from Finnish AMD protective haplotypes ($n = 27$) were performed in Graphpad Prism according to the distribution of data. Student's $t$ test was used for normally distributed data, and the Mann–Whitney $U$ test for skewed distributed data.

To verify that a higher complement activation capacity as tested in our complement assay on serum samples indeed translated to reduced complement activation, we tested what effects protective alleles in *CFH*, specifically at rs1410996 and/or p.Tyr402His, would have under our experimental conditions. *Risk* variants at both alleles would be expected to predispose to inflammation along with increased complement activation[13,30], whereas carriers of the respective protective alleles should have lower levels of complement activation with less inflammation, causing complement activation products. If this reasoning is correct, the latter should be reflected in a higher preserved complement activation capacity. Notably, this is exactly what we observed with our assay when we broke down the recall study participants into CFH protective haplotype carriers (n-n-0-0) and compared these against non-carriers (0-0-0-0). While our recall study was powered to test the functional effects of the $CFHR5_{fs}$ haplotype rather than that of the more common, but less impactful *CFH* variants, we still saw that *CFH* protective variant carriers showed a statistically significant increase in AP activation capacity ($p = 0.018$) (Supplementary Fig. 9). In AMD cases, both, CP ($p = 0.046$) and AP ($p = 0.034$), but not LP ($p = 0.26$) activation capacities were increased. These findings validate the hypothesis that *CFH* p.Tyr402His would show an effect in our assay and further support that indeed a downregulation of FHR-5 can be considered as the most promising strategy to prevent or treat AMD.

### FHR-5 western blot

For Western blot analysis of FHR-5 protein, serum samples from randomly selected recall study participants heterozygote or homozygote for $CFHR5_{fs}$ as well as non-carriers were mixed in gel loading buffer (LDS; Life Technologies) for non-reducing SDS-PAGE. SDS-PAGE was carried out with 4–12% Bis-Tris gradient gels (Thermo Fisher Scientific) at 165 Volt for 45 min. To visualize FHR-5, the proteins in the gel were electro-transferred to a nitrocellulose membrane (Thermo Fisher Scientific). The membranes were then incubated overnight at +4 °C with a goat polyclonal antibody against FHR-5 (R&D Systems, Cat.no. AF3845)[55] at a dilution of 1:5000. The membrane was washed with PBS/Tween and incubated for one hour at room temperature with HRP rabbit-anti-goat antibody (Jackson ImmunoResearch). Protein bands were visualized using electrochemi-luminescence. Purified FHR-5 was a gift from Mihaly Jozsi (Budapest, Hungary). Because of close structural similarity, the polyclonal antibody against FHR-5 also reacts with FH and structurally closely related other FHRs.

### UK Biobank retinal imaging quality control and association analyses

We extracted retinal layer thicknesses derived from UKB OCT scans as performed by Ko et al.[33] who had used version 1.6.1.1 of the topcon advanced boundary segmentation algorithm to segment 165,687 OCT scans from over 80,000 UKB participants. For our analysis, we excluded scans with an image quality score <45, as well as the 20% poorest quality scans based on remaining QC metrics which capture eye blinks, movement, and off-center images, among others (ILM indicator, macula center aline, macula center frame, max motion delta, max motion factor, min motion correlation, valid count; see refs. 33,38,56,57). Thresholds were defined across all scans from both eyes and participant visits, except for macula center aline, which was found to differ between left and right eyes. 85,036 scans from 56,874 participants passed our QC metrics. We focused on the photoreceptor layer (PS), and its two segments - inner (ELM-ISOS) and outer (ISOS-RPE) - as these layers were previously found to associate with AMD and a polygenic score for AMD risk[31,32]. Derived layer thicknesses were averaged across eyes for all participants. AMD cases were defined by having at least one entry for the ICD-10 code H35.3 (Degeneration of macula and posterior pole) in the in-patient hospital records as provided by UKB. The spherical equivalent for each eye was calculated from spherical power and cylindrical power and averaged across eyes. The effect of AMD case-control status on layer thickness was tested using linear regression, adjusting for age at scan, sex, spherical equivalent, center, and device ID. The four conditionally independent AMD-associated *CFH*/*CFHR5* regional variants under study were extracted by position from the UKB imputed genotypes and WES data in hail 0.2.116. All four variants were present in the UKB imputed genotypes, while three of the variants (rs1061170, rs139017763 and rs565457964) were also found in the WES data; thus, preference was given to the WES data for these variants. In the full UKB WES cohort, we identified 4,237 participants with additional alternate alleles at any of the four focal variants, indicating low-quality samples or haplotypes not present in Finns; these individuals were excluded from downstream analyses. For all genetic analyses, individuals were further restricted to unrelated individuals of White British ancestry without AMD ($n = 41,041$), and analyses were adjusted for the above covariates and the top 20 genetic principal components.

## FinnGen (DF12) ethics statement

FG study participants provided informed consent for biobank research based on the Finnish Biobank Act. Alternatively, separate research cohorts, collected before the Finnish Biobank Act came into effect (in September 2013) and the start of FinnGen (August 2017), were collected based on study-specific consents and later transferred to the Finnish biobanks after approval by Fimea (Finnish Medicines Agency), the National Supervisory Authority for Welfare and Health. Recruitment protocols followed the biobank protocols approved by Fimea. The Coordinating Ethics Committee of the Hospital District of Helsinki and Uusimaa (HUS) statement number for the FinnGen study is Nr HUS/990/2017.

The FinnGen study is approved by Finnish Institute for Health and Welfare (permit numbers: THL/2031/6.02.00/2017, THL/1101/5.05.00/2017, THL/341/6.02.00/2018, THL/2222/6.02.00/2018, THL/283/6.02.00/2019, THL/1721/5.05.00/2019 and THL/1524/5.05.00/2020), Digital and population data service agency (permit numbers: VRK43431/2017-3, VRK/6909/2018-3, VRK/4415/2019-3), the Social Insurance Institution (permit numbers: KELA 58/522/2017, KELA 131/522/2018, KELA 70/522/2019, KELA 98/522/2019, KELA 134/522/2019, KELA 138/522/2019, KELA 2/522/2020, KELA 16/522/2020), Findata permit numbers THL/2364/14.02/2020, THL/4055/14.06.00/2020, THL/3433/14.06.00/2020, THL/4432/14.06/2020, THL/5189/14.06/2020, THL/5894/14.06.00/2020, THL/6619/14.06.00/2020, THL/209/14.06.00/2021, THL/688/14.06.00/2021, THL/1284/14.06.00/2021, THL/1965/14.06.00/2021, THL/5546/14.02.00/2020, THL/2658/14.06.00/2021, THL/4235/14.06.00/2021, Statistics Finland (permit numbers: TK-53-1041-17 and TK/143/07.03.00/2020 (earlier TK-53-90-20) TK/1735/07.03.00/2021, TK/3112/07.03.00/2021) and Finnish Registry for Kidney Diseases permission/extract from the meeting minutes on 4th July 2019.

The Biobank Access Decisions for FinnGen samples and data utilized in FinnGen Data Freeze 12 include: THL Biobank BB2017_55, BB2017_111, BB2018_19, BB_2018_34, BB_2018_67, BB2018_71, BB2019_7, BB2019_8, BB2019_26, BB2020_1, BB2021_65, Finnish Red Cross Blood Service Biobank 7.12.2017, Helsinki Biobank HUS/359/2017, HUS/248/2020, HUS/430/2021 §28, §29, HUS/150/2022 §12, §13, §14, §15, §16, §17, §18, §23, §58, §59, HUS/128/2023 §18, Auria Biobank AB17-5154 and amendment #1 (August 17 2020) and amendments BB_2021-0140, BB_2021-0156 (August 26 2021, Feb 2 2022), BB_2021-0169, BB_2021-0179, BB_2021-0161, AB20-5926 and amendment #1 (April 23 2020) and it´s modifications (Sep 22 2021), BB_2022-0262, BB_2022-0256, Biobank Borealis of Northern Finland_2017_1013, 2021_5010, 2021_5010 Amendment, 2021_5018, 2021_5018 Amendment, 2021_5015, 2021_5015 Amendment, 2021_5015 Amendment_2, 2021_5023, 2021_5023 Amendment, 2021_5023 Amendment_2, 2021_5017, 2021_5017 Amendment, 2022_6001, 2022_6001 Amendment, 2022_6006 Amendment, 2022_6006 Amendment, 2022_6006 Amendment_2, BB22-0067, 2022_0262, 2022_0262 Amendment, Biobank of Eastern Finland 1186/2018 and amendment 22§/2020, 53§/2021, 13§/2022, 14§/2022, 15§/2022, 27§/2022, 28§/2022, 29§/2022, 33§/2022, 35§/2022, 36§/2022, 37§/2022, 39§/2022, 7§/2023, 32§/2023, 33§/2023, 34§/2023, 35§/2023, 36§/2023, 37§/2023, 38§/2023, 39§/2023, 40§/2023, 41§/2023, Finnish Clinical Biobank Tampere MH0004 and amendments (21.02.2020 & 06.10.2020), BB2021-0140 8§/2021, 9§/2021, §9/2022, §10/2022, §12/2022, 13§/2022, §20/2022, §21/2022, §22/2022, §23/2022, 28§/2022, 29§/2022, 30§/2022, 31§/2022, 32§/2022, 38§/2022, 40§/2022, 42§/2022, 1§/2023, Central Finland Biobank 1-2017, BB_2021-0161, BB_2021-0169, BB_2021-0179, BB_2021-0170, BB_2022-0256, BB_2022-0262, BB22-0067, Decision allowing to continue data processing until 31st Aug 2024 for projects: BB_2021-0179, BB22-0067,BB_2022-0262, BB_2021-0170, BB_2021-0164, BB_2021-0161, and BB_2021-0169, and Terveystalo Biobank STB 2018001 and amendment 25th Aug 2020, Finnish Hematological Registry and Clinical Biobank decision 18th June 2021, Arctic

biobank P0844: ARC_2021_1001. Analyses with UK Biobank data were conducted under the UKB project number 51766.

## Reporting summary

Further information on research design is available in the Nature Portfolio Reporting Summary linked to this article.

## Data availability

FG summary association results are available for download at https://www.finngen.fi/en/access_results and can be explored in a public results browser (https://r12.finngen.fi). All analyses in this manuscript which rely on variants that were directly interrogated through chip-based genotyping with the FG array or imputed rely on FG data freeze 12, which was publicly released in November 2024. Individual-level data from FG participants can be accessed by application to the Finnish Biobank Cooperative FinBB (https://finbb.fi/en/). Data and results from the FG recall study (including from proteomics analyses) are documented under the identifier "Complement Factor H refinement study in Finnish Biobanks" under the following link: https://app.fingenious.fi/cohorts. Consistent with the Finnish Biobank Act access requires approval by the respective biobanks and a material transfer agreement capturing the specific terms and conditions for access. A contact email address is provided on the website. Replies should be expected within a few days. Links to further source datasets and results: FG summary association results: https://r12.finngen.fi. FG individual-level results: https://finbb.fi/en/ (via application according to Finnish Biobank Act). 1KG long-read WGS data and HPRC haplotypes: https://humanpangenome.org/data/. 1KG phased short-read WGS data: https://www.internationalgenome.org/data. UKB proteomics data: Category #1839 (https://biobank.ndph.ox.ac.uk/ukb/label.cgi?id=1839). UKB WES data: Category #170 (https://biobank.ndph.ox.ac.uk/ukb/label.cgi?id=170). UKB OCT data: Category #100016 (https://biobank.ndph.ox.ac.uk/ukb/label.cgi?id=100016). UKB Retinal layer thickness: Project #2112 (https://biobank.ndph.ox.ac.uk/ukb/app.cgi?id=2112).

## Code availability

FG data analysis pipelines are freely available from https://github.com/FINNGEN/. The FinnGen Handbook, https://finngen.gitbook.io/documentation/, contains a detailed description of data production and analysis, including code used to run the analyses described in this manuscript.

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

## Acknowledgements

We thank all the participants, contributors, and researchers of FinnGen and its participating biobanks for making data or samples available for this study. We are grateful to Marcel Messing for his help with conducting complement analyses. The FinnGen project is funded by two grants from Business Finland (HUS 4685/31/2016 and UH 4386/31/2016) and the following industry partners: AbbVie Inc., AstraZeneca UK Ltd, Biogen MA Inc., Bristol Myers Squibb (and Celgene Corporation & Celgene International II Sàrl), Genentech Inc., Merck Sharp & Dohme LCC, Pfizer Inc., GlaxoSmithKline Intellectual Property Development Ltd., Sanofi US Services Inc., Maze Therapeutics Inc., Janssen Biotech Inc, Novartis AG, and Boehringer Ingelheim International GmbH. Following biobanks are acknowledged for delivering biobank samples to FinnGen: Auria Biobank (www.auria.fi/biopankki), THL Biobank (www.thl.fi/biobank), Helsinki Biobank (www.helsinginbiopankki.fi), Biobank Borealis of Northern Finland (https://www.ppshp.fi/Tutkimus-ja-opetus/Biopankki/Pages/Biobank-Borealis-briefly-in-English.aspx), Finnish Clinical Biobank Tampere (www.tays.fi/en-US/Research_and_development/Finnish_Clinical_Biobank_Tampere), Biobank of Eastern Finland (www.ita-suomenbiopankki.fi/en), Central Finland Biobank (www.ksshp.fi/fi-FI/Potilaalle/Biopankki), Finnish Red Cross Blood Service Biobank (www.veripalvelu.fi/verenluovutus/biopankkitoiminta), Terveystalo Biobank (www.terveystalo.com/fi/Yritystietoa/Terveystalo-Biopankki/Biopankki/) and Arctic Biobank (https://www.oulu.fi/en/university/faculties-and-units/faculty-medicine/northern-finland-birth-cohorts-and-arctic-biobank). All Finnish Biobanks are members of BBMRI.fi infrastructure (https://www.bbmri-eric.eu/national-nodes/finland/). Finnish Biobank Cooperative -FINBB (https://finbb.fi/) is the coordinator of BBMRI-ERIC operations in Finland. The Finnish biobank data can be accessed through the Fingenious® services (https://site.fingenious.fi/en/) managed by FINBB. Further support for the study came from Biogen Inc, the Finnish Institute of Molecular Medicine (FIMM), the Academy of Finland (Project #336411), the Sigrid Jusélius Foundation (#4708373), Special State Subsidy for Health Research at Helsinki University Hospital (VTR-funding, TYH2023322), the National Institutes of Health grants MH115957 and HG011450, and the Health + Life Science Alliance Heidelberg Mannheim.

## Author contributions

Conceptualisation and design: M.P.R., S.L., M.D., H.R. Methodology: M.P.R., S.L., E.N., T.S., T.R., S.M., M.D., H.R. Analysis: M.P.R., S.L., E.N., T.S., T.R., Z.Z., P.B.P., S.D.I., E.A., E.C., Y.O., M.K., M.D., H.R. Experimental work: E.N., E.A., E.C., S.M. FinnGen/biobank protocols and analysis: M.P.R., H.M.L., J.M., M.D., H.R., FinnGen. Supervision and/or funding: M.K., M.T., J.K., K.C., S.M., M.D., H.R., FinnGen. Writing: M.P.R., S.L., M.D., H.R. All authors critically reviewed the manuscript.

## Competing interests

S.L., Y.O., H.M.L., K.C., and H.R. were employees at Biogen during data generation for this study. S.L. is an employee of Bristol-Myers Squibb. Y.O. and K.C. are employees of Johnson & Johnson. H.M.L. is an employee of Moderna. J.M. is an employee of FinBB. M.D. is a co-founder of Maze Therapeutics. TS and HR are employees at insitro Inc. The remaining authors declare no competing interests.

## Additional information

## FinnGen

Mary Pat Reeve [ORCID]1,2,3,12, Stephanie Loomis4,12, Zhili Zheng [ORCID]2,3, Pietro DELLA BRIOTTA PAROLO2,3, Johanna Mäkelä11, Mitja Kurki1,2,3, Mark J. Daly [ORCID]1,2,3,13 ✉ & Heiko Runz [ORCID]1,4,6,7,13 ✉

