## [Transparent Peer Review file · Nature Communications]

Loss of CFHR5 function reduces the risk for age-related macular degeneration

Corresponding Author: Dr Heiko Runz

Version 0:

Reviewer comments:

Reviewer #1

(Remarks to the Author)

The authors present findings of a genome-wide association study (GWAS) utilizing genome and national electronic health register data from FinnGen (freeze 12) including 12,495 cases diagnosed with age-related macular degeneration (AMD) and 461,686 controls. They successfully replicate 17 of the 34 known AMD-associated loci previously identified by Fritsche et al. in 2016 and report six novel AMD loci. The study particularly highlights the complement factor H (CFH) region on chromosome 1q, which includes CFH and five related genes (CFHR1-5). Among the major new discoveries are two coding variants in CFHR5 - p.Glu163insAA and p.Gly278Ser - found to confer a dose-dependent protection against AMD enriched in the Finnish population. Additionally, the study observes that the CFHR5 frameshift variant p.Glu163insAA significantly reduces serum levels of FHR-2, FHR-4, and FHR-5, alongside an increased activity of the classical and alternative, but not the lectin complement pathways, pathway.

The study is important and adds novel data to further understand the role of the CFH/CFHR region to AMD susceptibility. Overall, the manuscript is articulate and informative. However, it could be improved by tuning some of the more expansive conclusions which at times appear to overinterpret the data. Also, there are several inaccuracies/missing details that make it difficult for the reader to fully understand the figures, extended data figures, tables, and supplementary tables.

Specifically, the following points should be addressed by the authors:

1. Lines 84-85: the authors state that their data show “genetic loss of CFHR5 function is associated with a reduced risk for AMD irrespective of other variation at the CFH locus”. This, however contradicts their statement in the discussion section “our recall study from Finnish samples cannot explain whether loss of FHR-5 function would also raise complement activity in a setting where FH is unimpaired, since all carriers of CFHR5fs and CFHR5 p.Gly278Ser in the FG cohort also carried at least one of the two AMD protective alleles in CFH” which needs to be clarified (line 448-451). The reviewer agrees with the latter statement and would recommend drawing a precise conclusion based on the results given.
2. Lines 98-99 and Supplementary Table 1: Compared to the findings by Fritsche et al. 2016 the FinnGen analysis replicates only 17 of 34 significantly associated AMD loci although the statistical power should be comparable in the two studies. Maybe the authors would like to discuss possible reasons for this noticeable finding (e.g. effects of genetic drift, AMD cases in FinnGen is a mixture of early-stage and late-stage cases with different risk profiles, etc.). Also, it would be highly informative to include the FinnGen GWAS data of all Fritsche et al. 2016 AMD loci in Supplementary Table 1.
3. Line 100: it is unclear what is meant by “three out of five regions”.
4. Lines 101-103, lines 104-112: Functionality cannot be deduced from statistical findings (e.g. see the authors’ candidate selection of LIPC over ALDH1A2 or ARMS2 over HTRA1).
5. Line 106: Statements like “previously described” should be referenced.
6. Lines 108-109, line 148: How is the enrichment in the Finnish population defined? What is the MAF in other populations? A comment on the prevalence of AMD in Finland and a corresponding setting of these results in context would be appreciated.
7. Line 114: The header is inaccurate and is not defining the content of the following paragraph. The phenome-wide scan which is supposedly a central part of this paragraph is not addressed. Also, this paragraph mixes results like “major GWAS findings” and at the same time provides results of the CFH finemapping by addressing independent CFH signals and a specific CFHR3 variant. Maybe the authors would like to restructure the results to provide a clear separation of major GWAS findings and finemapping results.
8. Line 124: What is the impact of the CFHR5 fs variant on protein level?
9. Lines 129-132: The comment “Such lack of evident signs for adverse events in humans contributes to make therapeutic reduction of CFHR5 or its products a potentially compelling strategy for the prevention or treatment of AMD, which motivated us to further de-risk this putative drug target through refined genetic and functional studies” seems too strong considering

that the PheWAS was conducted in a single dataset without replication for a gene/protein involved in the complement pathway. Possibly a replication in UK Biobank data could strengthen this statement.

10. Lines 156-158: The statement "Notably, this fourth signal also included CFHR5fs ($r^2=0.91$) that, like CFHR5 p.Gly278Ser, was associated with protection from AMD, indicating a potential allelic series in CFHR5 that could be leveraged to develop targeted therapies" is again an overinterpretation of data, for two reasons: 1. not only CFHR5 variants are part of the fourth signal and 2. it is unclear whether the protective effect is related to the CFHR5 variant or to another closely linked variant. Notably, the data displayed in Table 1 do not support a protective effect of the variant as the allele frequency is higher in cases than controls.

11. Lines 190-193: It is widely accepted that coding variants are of primary interest for causal variants, although they need to be validated by additional functional studies. Of course, a non-coding variant can also be an effective variant. Specifically, a non-coding variant may have regulatory effects on gene expression of any of the CFHR1-5 genes. Such a situation was not addressed in this study. Thus, the statement "Because these are common deletion variants, this provides little support for the hypothesis that genetic variants impacting CFHR1, CFHR3 and CFHR4 contribute substantially to the strong association between the CFH region and AMD - which our results suggest are primarily explained by variants in CFH and CFHR5 genes" seems too strong and should be rephrased.

12. Line 236: The term "pleiotropy" is used to refer to the expression of different phenotypic traits caused by a single gene. This does not appear to be the correct terminus technicus in the given context.

13. Lines 248-250: The statement "Taken together, our base-level interrogation of the CFH region excludes that the reduced risk for AMD in CFHR5 frameshift and missense variant carriers is conveyed by local SVs or concealed single base pair changes other than those impacting CFHR5" is incorrect as it ignores that the CFHR5 variants were detected as part of an association signal which per se is no proof of causal protective variants in CFHR5.

14. Lines 306-308: Again, the statement "This demonstrates that the protective effects of CFHR5fs on serum protein levels are independent of and add to that from other protective variation at the CFH locus" is an overinterpretation of the data. No independence was shown in the results. Notably, Supplementary Tables 12 and 13 indicate rather the opposite.

15. Lines 329-366 and Extended Data Figure 7: The results described in this paragraph are partially not in line with the data as given in the figures provided. For example, the postulated "...intermediate effects between homozygotes and non-carriers" are not visible in Extended Data Figure 7. In fact, the values of heterozygotes look similar to non-carriers. Also, in Figure 5c there is no significant difference between individuals carrying one CFHR5fs allele compared to individuals carrying no CFHR5fs allele (as described in the text in lines 347-352).

16. Lines 357-365: Although the authors state that "...results from these functional analyses demonstrate that genetic loss of CFHR5 function is independently associated with increased activation capacity of the complement system to confer protection from AMD", independence was not shown in the performed functional analyses.

17. Lines 195-250: The frequency of CFHR5 variants is compared between Finns and non-Finnish individuals, however the latter group includes several related individuals. This results in biased allele frequencies between the two groups (see also Supplementary Table 4). In this context, it is unclear whether individuals included in investigating CFHR serum levels are related or unrelated.

18. Lines 252-278 and Supplementary Table 9: The distribution of haplotypes for the two major independent signals should be summarized and mentioned in the main text. This should also clarify if the genotypes of the two major signals are evenly distributed between CFHR5fs n/c, het and hom and the different AMD groups.

19. Lines 276-278 and Supplementary Table 10: In the text it is stated that no significant effect of FHR1 was observed for a p-value cutoff of 0.05. In Supplementary Table 10, however, the respective p-value is 0.03 and therefore below the threshold. The same is true for the AMD-Status. Applying the same p-value cutoff would result in significant results for FHR-2 and FHR-5. This would indicate that the AMD status has an effect on FHR-2 and FHR-5 serum levels which is important as the distribution of hetero- and homozygous carriers of the frameshift variant are not evenly distributed between AMD cases and controls.

20. Line 335-345 and Figure 5b: Samples sizes of the three different genotypes should be given.

21. Supplementary Table 12: Data indicate that the major independent signals in the CFH locus also influence serum levels. Unfortunately, the information in the Methods is insufficient to judge if adjustment for independent signals which occur in the respective individuals was performed in the analysis. Also, FHR-1 reveals a significant result for the CFHR5 frameshift variant, while FHR-4 fails to reach significance. In fact, this is consistent with Supplementary Table 13, where an effect for FGAMD1 and FGAMD2 was observed for several FH and FHR serum levels.

22. Reference 17 is highly relevant to the current study and should be critically included in the Discussion section.

23. Supplementary Table 2: Confusing mix of data from Fritsche et al. 2016 and own data. This should be resolved and presented in a structured way to make the point clear to the reader.

24. Supplementary Table 4: Abbreviations should be explained. Heterozygous alleles are given in the form 0/1 or 1/0. This should be harmonized.

25. Supplementary Table 6: unclear if columns indicated with AC refer to individuals/haplotypes. If so, why are several individuals/haplotypes included more than once? For example, AC=11 occurs four times and always with different variants.

26. Supplementary Table 7: Terms "FIN%FIN" and "FIN%ALL" should be clarified.

27. Supplementary Table 8: Heading needs to be changed from Supplementary Table 10 to Supplementary Table 8.

28. Methods: It is unclear how UK Biobank data were analyzed and how the burden for coding variants was determined.

29. Line 562: Nomenclature for the HTRA1 variant 10:122461685:G:GTCCCT should read 10:122461685:C:TCCT.

30. Figure 1: In Fig. 1a signal 1.5 is missing. Also, the missense variant Tyr402His should be located in an exon of CFH. Why are some genes highlighted in green?

31. The terms FGAMD3 and FGAMD4 are mixed in the text with the descriptions missense and frameshift variant, respectively. Maybe it would help the reader to refer to a single terminology.

32. Figure 2a: It is unclear why Fig. 2a annotates a SNP twice with "CFHR5p.Glu163insA".

33. Table 1: AMD cases add up to 200 (not 199 as indicated in the last column).

34. Legend to Figure 5: It is unclear why a student's t-test was performed and for which analysis? The data show no two

group comparisons.

Reviewer #2

(Remarks to the Author)

Reviewer #3

(Remarks to the Author)

This study describes a GWAS for AMD in FinnGen, a unique founder population that enables identification of rare variants and can pinpoint causal genes. Of particular interest are a rare coding variant in the HTRA1 gene suggesting a causal role of HTRA1 in AMD, in addition to rare frameshift variant in CFHR5 at the CFH locus. Circulating protein levels of 6000 proteins, including FH and FHR-1-5, were measured using the Somascan platform. Notably, not only CFHR5, but also CFHR2 and CFHR4 were decreased in carriers of the CFHR5 frameshift variant while complement activity levels were increased.

The CFHR5 frameshift variant was found to be protective for AMD, and is associated with reduced FHR-5 levels using the Somascan platform. Surprisingly complement activity (AP and CP) levels are increased in carriers of the CFHR5 frameshift variants (Figure 5). Increased complement activation levels have consistently been described to lead to increased risk for AMD, therefore, this finding is highly unexpected. The conclusion "Our findings ... propose therapeutic downregulation of FHR-5 as a promising strategy for prevention or treatment of AMD" is therefore incorrect. A therapeutic strategy for downregulating FHR-5 would only be warranted if genetic loss of CFHR5 would be associated with decreased risk of AMD and decreased complement activity levels. This questions the validity of the conclusions drawn from the AP and CP activity assays used in this study; further investigation using other commonly used and more robust assays to assess complement activity is warranted (e.g. C3b degradation and/or C3d/C3 ratio), in addition to in vitro studies to assess the isolated effect of CFHR5 variants using recombinantly expressed CFHR5 protein on complement activity. Another approach to assess the validity of the AP and CP activity assays is to evaluate the effect of the CFH Y402H risk variant on complement activity, as these levels are expected to be increased by this risk allele.

Notably, the CFHR5 frameshift variant was found to be protective for AMD, while a CFHR5 missense variant (FG-AMD3) was found to be associated with increased AMD risk (Figure 4). This further complicates the conclusion that loss of CFHR5 is associated with decreased risk of AMD; the CFHR5 frameshift variant supports this conclusion but the CFHR5 missense variant does not.

The authors refrained to evaluate the effect of the rs800292 (V62I) variant in the CFH gene leading to a valine to isoleucine change at amino acid position 62 of the CFH protein, which has been postulated to be protective for AMD by prominent researchers in the AMD field. An assessment of common risk and protective variants at the CFH locus, in particular the rs800292 coding (V62I) variant in the CFH gene, on CFH and CFHR protein levels may help clarify an ongoing debate related to the studies published by Zouache et al (Nat Commun 2024 Jan 10;15(1):443) and Cipriani et al (Nat Commun. 2020 Feb 7;11(1):778.). How does the V62I variant relate to the evaluated CFH haplotypes; is this variant independently associated with AMD, and what is the effect of this variant on CFHR protein levels and complement activity?

Reviewer #4

(Remarks to the Author)

Reeve et al. demonstrate through fine mapping and conditional analyses that four independent haplotypes in the CFH genomic region on chromosome 1 confer protection against age-related macular degeneration (AMD) in Finns, as shown in the large-scale FinnGen genetic cohort. Two of these haplotypes, which reflect deleterious variants within the CFHR5 gene, are enriched in the Finnish population compared to another European cohort (UK Biobank data). These haplotypes are associated with lower levels of CFHR5 (FHR-5), as well as CFHR2 (FHR-2) and CFHR4 (FHR-4), and correlate with increased activity of relevant complement system pathways. The authors had previously reported a frameshift variant within CFHR5 (Glu163insAA, rs565457964) that provides protection against AMD, meaning the protective effect conferred by the new haplotypes is not entirely novel. Nevertheless, the identification of these four independent genetic signals within the complex AMD linked region on chromosome 1 represents new information for the field.

Main Point of Criticism:

The main issue with this manuscript is its lack of novelty. As mentioned, the authors previously published the findings of a frameshift allele in CFHR5 that was protective for AMD. In addition, other researchers published work in Nature Communications (PMID: 35697682), that documented several aspects of the relationship between the protein CFHR5 (aka FHR-5) and AMD. This included its involvement in late-stage AMD, its potential to predict advanced AMD, and its causal relationship with advanced AMD, as demonstrated through two-sample Mendelian randomization analysis. This work was unfortunately not cited in this manuscript. These two reports have already established a link between CFHR5 and AMD, and

as such there is little novelty to Reeve et al.'s findings.

The main claimed point of the manuscript is the clear identification of CFHR5 as the culprit in AMD. Yet the protective CFHR5 haplotypes were not only associated with lower CFHR5 (FHR-5), but CFHR2 (FHR-2) and CFHR4 (FHR-4) as well. In other words, this reflects molecular pleiotropy that brings into question the authors main claim. The authors seem to be somewhat selective when discussing this point. While they frequently suggest that targeting CFHR5 (FHR-5) could be a novel therapeutic strategy for AMD, their results also support a strategy of lowering CFHR2 (FHR-2) and CFHR4 (FHR-4) levels as well. In fact, the previously mentioned missing citation (Nature Communications, PMID: 35697682) found that the CFH variant rs10922109_A allele, which confers protection against AMD, was associated with lower levels of CFHR4 (FHR-4) protein. This aspect of their work should have been acknowledged, as its omission undermines the impact of their findings.

Additional Points of Criticism:

FinnGen, a public resource, is one of the many prominent modern genetic cohorts utilizing large datasets to enhance statistical power in detecting rare variants associated with disease. An important limiting factor that cannot be solely addressed by increasing sample size, however, is the quality and depth of phenotype annotation and measurements within the cohort. Comprehensive phenotypic data provides more precise insights, enabling researchers to distinguish between meaningful genetic signals and potential confounders. It is unclear from the manuscript to what extent the AMD population in this study is characterized in terms of different AMD types, stages, and progression, including the transition from early to late stages, such as geographic atrophy or the more severe neovascular form of AMD. Associating the different haplotypes with these measures would be of significant interest to the field. A fuller discussion of this point would have been appreciated.

Furthermore, the relationship between protein levels and disease risk is not always symmetric: while an increase in the level of a protein that raises disease risk might imply that inhibiting or reducing its activity could lower the risk, this is not always the case due to factors such as compensatory mechanisms and natural feedback loops. Therefore, demonstrating that elevated CFHR5 levels increase the risk of AMD is just as novel as showing that reduced levels offer protection, and may indicate that in this instance, symmetry might apply. A better discussion of this in the context of all prior published work would have been a nice addition too.

Page 13, lines 451-452. "With the emergence of both sets of variants on the same haplotype, evolution may have found a mechanism to re-adjust the balance between FH and FHR-5 in a most favorable way to prevent disease". How does this sentence make any sense, given that negative (purifying) or positive selection forces typically act to preserve or enhance reproductive fitness, yet AMD (assuming this is the disease referred to) primarily affects older adults, who are in the post-reproductive age?

A related, though potentially less relevant question considering the rarity of the protective CFHR5 haplotypes: given that these haplotypes are enriched in Finns, likely due to population bottleneck and/or genetic drift, are there any epidemiological studies that show whether the prevalence or incidence of AMD differs between Finns and other European populations?

While revising the manuscript by Reeve et al., the largest multi-ancestry GWAS meta-analysis of genetic risk factors for AMD was published (Nature Genetics, PMID: 39623103), identifying many new loci associated with AMD risk. Reeve et al. report that 23 risk loci (including 6 new loci) for AMD have been identified through the FinnGen study. Considering this, the authors should alter their manuscript to clarify the number of new loci identified in FinnGen in comparison to this more comprehensive genetic study of AMD risk.

Why do the authors use different annotations for the gene (CFHR5) and its protein product (FHR-5)? The protein annotation seems to be based on historical usage and may no longer be up to date.

Version 1:

Reviewer comments:

Reviewer #1

(Remarks to the Author)

The authors have undertaken a comprehensive revision of the manuscript, thoroughly addressing all points raised by Reviewer 1. These revisions have significantly improved the readability and conclusiveness of the study. Notably, the authors have incorporated extended data tables and figures and have substantially expanded the Discussion section, including additional references. Furthermore, their refined interpretation of the data is acknowledged and appreciated. These improvements enhance the overall rigor and comprehensiveness of the study.

Two points remain that the authors may consider addressing:

1. AMD Prevalence in Finns (Reviewer 1, Former Point 6)

It would be beneficial to include information on AMD prevalence in Finns within the manuscript itself rather than solely in the rebuttal letter. Providing this context would enhance the interpretability of the findings for readers.

2. Allele Frequency Comparison and Related Individuals (Reviewer 1, Former Point 17)

The removal of fold-enrichment estimates for Finns versus non-Finns is noted and appreciated. However, a key concern remains. While our initial critique may have been misinterpreted, comparing allele frequencies between a cohort that includes related individuals and one that does not may introduce bias, particularly given the limited sample size. A more rigorous approach would involve excluding related individuals from the 1000 Genomes (1KG) cohort by filtering based on family ID and recalculating allele frequencies. While this adjustment would reduce the sample size, it would yield more comparable and less biased allele frequency estimates.

Finally, we want to emphasize that the evaluation of revisions made in response to points raised by Reviewers 3 and 4 is left to their discretion and expertise, as these aspects were not assessed by Reviewer 1.

Reviewer #2

(Remarks to the Author)

Reviewer #3

(Remarks to the Author)

Thank you for addressing the concerns of the reviewers, and for clarifying several areas that were unclear. Although the manuscript has improved with the edits, I do not see much novelty of the findings described in the manuscript. As also pointed out by reviewer 4, the authors describe CFHR5 variants that have been previously described to have a protective effect on AMD, and that have previously been shown to reduce FHR-5 levels (Emilsson et al 2022, Nat Commun 13:2401; Lores-Motta et al 2021, AJHG 108:1367).

The authors have also not attempted to clarify the ongoing debate related to the studies published by Zouache et al (Nat Commun 2024 Jan 10;15(1):443) and Cipriani et al (Nat Commun. 2020 Feb 7;11(1):778.), while this could have been a strength of this manuscript.

Reviewer #4

(Remarks to the Author)

I believe the authors have addressed my comments comprehensively, and I have no further remarks

Version 2:

Reviewer comments:

Reviewer #1

(Remarks to the Author)

In the re-revision the authors have now addressed all concerns of reviewer #1. Thanks to the authors for their responsiveness.

Reviewer #2

(Remarks to the Author)

Point-by-point response to reviewers:

Reviewer #1 (Remarks to the Author):

The authors present findings of a genome-wide association study (GWAS) utilizing genome and national electronic health register data from FinnGen (freeze 12) including 12,495 cases diagnosed with age-related macular degeneration (AMD) and 461,686 controls. They successfully replicate 17 of the 34 known AMD-associated loci previously identified by Fritsche et al. in 2016 and report six novel AMD loci. The study particularly highlights the complement factor H (CFH) region on chromosome 1q, which includes CFH and five related genes (CFHR1-5). Among the major new discoveries are two coding variants in CFHR5 - p.Glu163insAA and p.Gly278Ser - found to confer a dose-dependent protection against AMD enriched in the Finnish population. Additionally, the study observes that the CFHR5 frameshift variant p.Glu163insAA significantly reduces serum levels of FHR-2, FHR-4, and FHR-5, alongside an increased activity of the classical and alternative, but not the lectin complement pathways, pathway.

The study is important and adds novel data to further understand the role of the CFH/CFHR region to AMD susceptibility. Overall, the manuscript is articulate and informative. However, it could be improved by tuning some of the more expansive conclusions which at times appear to overinterpret the data. Also, there are several inaccuracies/missing details that make it difficult for the reader to fully understand the figures, extended data figures, tables, and supplementary tables.

Specifically, the following points should be addressed by the authors:

1. Lines 84-85: the authors state that their data show “genetic loss of CFHR5 function is associated with a reduced risk for AMD irrespective of other variation at the CFH locus”. This, however contradicts their statement in the discussion section “our recall study from Finnish samples cannot explain whether loss of FHR-5 function would also raise complement activity in a setting where FH is unimpaired, since all carriers of CFHR5fs and CFHR5 p.Gly278Ser in the FG cohort also carried at least one of the two AMD protective alleles in CFH” which needs to be clarified (line 448-451). The reviewer agrees with the latter statement and would recommend drawing a precise conclusion based on the results given.

We thank the reviewer for their positive feedback and highlighting opportunities to add further precision to the communication of our results, which has substantially improved the clarity of our manuscript. We have now carefully revised our manuscript to further reduce the risk that our findings could be overinterpreted and apologize for any confusion. For instance, the above mentioned sentence now reads: “We show that carrying a haplotype that leads to genetic loss of CFHR5 function is associated with a reduced risk for AMD independent of other evident association signals at the CFH locus“. Also in other sections of the manuscript we now refer to the CFHR5 loss-of-function *haplotypes* rather than the distinct CFHR5 coding variants in isolation.

2. Lines 98-99 and Supplementary Table 1: Compared to the findings by Fritsche et al. 2016 the FinnGen analysis replicates only 17 of 34 significantly associated AMD loci although the statistical power should be comparable in the two studies. Maybe the authors would like to discuss possible reasons for this noticeable finding (e.g. effects of genetic drift, AMD cases in FinnGen is a mixture

of early-stage and late-stage cases with different risk profiles, etc.). Also, it would be highly informative to include the FinnGen GWAS data of all Fritsche et al. 2016 AMD loci in Supplementary Table 1.

Fritsche et al., 2016 included 16,144 AMD cases and 17,832 controls, while our FinnGen (FG) DF12 AMD GWAS included 12,495 AMD cases and 461,686 controls. We have detailed in previous publications (e.g., Sun et al., 2022; Kurki et al., 2023) how the unique Finnish haplotype structure with at times multiple-fold different allele frequencies than in other Europeans can boost the power for association testing for some GWAS loci, while falling short in providing a similar association power for others. For the revised manuscript, we updated our comparisons based on the recently published multi-ancestry meta-GWAS by Gorman et al. (61,248 AMD cases and 364,472 controls). As we highlight in **Results**, of the 62 loci identified as associated with AMD in that study, 19 loci met genome-wide significance criteria in FG, 31 replicated at nominal significance ($p < 0.05$) and 11 did not replicate (although 10 of the 11 appear to be in the same direction as in Gorman et al., with one locus not having been included in the FG analysis set). We have annotated the FG association results with information from Fritsche et al. and Gorman et al. in a revised **Supplementary Table 1** and added a new **Supplementary Table 2** that provides FG association results for all loci described by Gorman et al.

As we highlight in the manuscript, all FinnGen DF12 results, including for the phenotype studied in our manuscript (H7_AMD), are publicly available for browsing (<http://r12.finnngen.fi>) and download of summary statistics (https://www.finnngen.fi/en/access_results).

3. Line 100: it is unclear what is meant by “three out of five regions”.

The corresponding sentence now reads: “*Systematic finemapping and conditional analyses prioritized likely causal variants within three out of five FG GWAS regions with multiple apparently independent signals.*”

4. Lines 101-103, lines 104-112: Functionality cannot be deduced from statistical findings (e.g. see the authors’ candidate selection of LIPC over ALDH1A2 or ARMS2 over HTRA1).

We have removed our hypothesis that based on our finding of a conditionally independent intronic insertion *LIPC* might be considered as more likely than *ALDH1A2* to drive the respective GWAS association signal. However, we would like to point out that the lead variant at this locus in Gorman et al. (rs2414577; 15:58680638:T:C) now also locates to *LIPC* as the nearest coding gene. For the *HTRA1/ARMS2* locus, we report a conditionally independent rare insertion in the coding region of *HTRA1* that has likely been missed in previous attempts to deconvolute this GWAS signal. Through a look up in primary sequencing data from gnomAD we found that this variant had been mis-annotated as two separate frameshift variants (see Methods). Instead, it is derived from a single mutational event that introduces a serine residue into a conserved string of leucine residues near the start of the *HTRA1* protein-coding sequence, which is highly likely to have consequences on protein function. In our manuscript, we continue to abstain from any comments

on likely causalities or functionality at this locus, but given the high interest in the field we have decided not to modify the description of our findings.

5. Line 106: Statements like “previously described” should be referenced.

We are now referencing the original publication reporting an AMD association with the *ARMS2* p.Ala69Ser variant: *Rivera, A. et al. Hypothetical LOC387715 is a second major susceptibility gene for age-related macular degeneration, contributing independently of complement factor H to disease risk. Hum Mol Genet 14, 3227-3236 (2005).*

6. Lines 108-109, line 148: How is the enrichment in the Finnish population defined? What is the MAF in other populations? A comment on the prevalence of AMD in Finland and a corresponding setting of these results in context would be appreciated.

We have detailed how variant enrichment in Finns relative to non-Finns is determined in previous publications (e.g., Sun et al., 2022; Kurki et al., 2023). In brief, enrichment scores reflect the ratio of allele frequencies (AF) in whole genome and whole exome sequencing reference datasets from Finnish individuals (SISu v3 imputation panel, gnomAD exome v2.1 panel from Finnish participants, and Finnish exome collection; encompassing approximately 8,700 individuals) to that of non-Finnish European (NFE) individuals in gnomAD v2.1 (~140,000 samples). In the latest version of gnomAD (v4.1.0; n=807,162 samples) the *HTRA1* insertion in question was observed in 295 of 30,686 Finnish alleles (AF=0.009614) and in 242 of 1,010,366 NFE alleles (AF=0.0002395), reflecting an ~40-fold enrichment. In the manuscript we have substituted the previously reported AF with this enrichment score. For prevalence in other populations, see https://gnomad.broadinstitute.org/variant/10-122461685-G-GT?dataset=gnomad_r4.

Based on FinRegistry data from health records of ~5 million Finns (see: https://risteys.finngen.fi/endpoints/H7_AMD), the unadjusted period prevalence of broadly defined AMD is 1.64% in the overall Finnish population and 2.48% among the (on average older) FinnGen population. This prevalence rises steeply with older age, and among 90-year-olds in Finland, 15.1% of women and 10.0% of men have received a registry-based diagnosis of AMD. Consistent with epidemiological information from other countries, the median age at diagnosis in Finns is 79.03 years and 76.26 years in FinnGen. Thus, the prevalence of broadly-defined AMD in Finland does not seem to differ substantially from the prevalence of progressed AMD in other European populations with comparable age distributions.

7. Line 114: The header is inaccurate and is not defining the content of the following paragraph. The phenome-wide scan which is supposedly a central part of this paragraph is not addressed. Also, this paragraph mixes results like “major GWAS findings” and at the same time provides results of the CFH finemapping by addressing independent CFH signals and a specific CFHR3 variant. Maybe the authors would like to restructure the results to provide a clear separation of major GWAS findings and finemapping results.

We have now better separated GWAS and finemapping results into two separate paragraphs. For this, we have now merged the paragraph referred to with the preceding paragraph on FG GWAS findings.

8. Line 124: What is the impact of the CFHR5 fs variant on protein level?

A Western Blot that independently confirms that serum samples of FHR-5 are reduced in heterozygous *CFHR5_{fs}* carriers and presumably absent in individuals that are homozygous for *CFHR5_{fs}* has been added as a new **Extended Data Figure 7**.

9. Lines 129-132: The comment “Such lack of evident signs for adverse events in humans contributes to make therapeutic reduction of CFHR5 or its products a potentially compelling strategy for the prevention or treatment of AMD, which motivated us to further de-risk this putative drug target through refined genetic and functional studies” seems too strong considering that the PheWAS was conducted in a single dataset without replication for a gene/protein involved in the complement pathway. Possibly a replication in UK Biobank data could strengthen this statement.

We have now modified this sentence as follows: “*The strong GWAS signal in the absence of evident signs for adverse events in FinnGen motivated us to further de-risk CFHR5 through refined genetic and functional studies for its potential as a drug target to prevent or treat AMD.*”

Due to its ~10-fold lower carrier frequency in non-Finnish Europeans (AF ~0.4%) relative to Finns (AF ~4%), a PheWAS for *CFHR5_{fs}* (chr1:196994128:C:CAA) against 4,529 phenotypes ascertained from up to 394,841 UK Biobank (UKB) participants (which include only 234 heterozygous *CFHR5_{fs}* carriers) in the genebase resource (<https://app.genebase.org>; Karczewski et al., 2022) does not yield results of genome-wide significance. However, a meta-PheWAS across FG DF12 (n=500,349), UKB (n=420,531) and the Million Veteran Project (MVP) (n=449,042 Europeans, 121,177 Africans, 59,048 Hispanics) cohorts further consolidates that *CFHR5_{fs}* is associated with “Degeneration of macula and posterior pole” (p=1.4e⁻³⁴; beta=-0.362) but none of the 330 other tested disease endpoints (<https://mvp-ukbb.finnngen.fi/variant/1-196994128-C-CAA>). Since these results from a larger cross-ancestry analysis are based on a smaller number of phenotypes tested and do not substantially change our results in FG alone we have decided not to discuss these data in the manuscript.

10. Lines 156-158: The statement “Notably, this fourth signal also included CFHR5fs (r²=0.91) that, like CFHR5 p.Gly278Ser, was associated with protection from AMD, indicating a potential allelic series in CFHR5 that could be leveraged to develop targeted therapies” is again an overinterpretation of data, for two reasons: 1. not only CFHR5 variants are part of the fourth signal and 2. it is unclear whether the protective effect is related to the CFHR5 variant or to another closely linked variant. Notably, the data displayed in Table 1 do not support a protective effect of the variant as the allele frequency is higher in cases than controls.

Thanks to the reviewer we realized that the table of conditional analyses in Figure 1 had been misformatted so that two independent signals (1.5 and *CFHR5_{fs}*) appeared accidentally as

merged. We have reformatted **Figure 1** and **Supplementary Table 3** to clarify the independent effects of the associations described. We apologize for the formatting error which indeed made it appear that the frameshift variant was more common in cases than controls. In fact, in both univariate and conditional analyses, the frameshift is significantly less often seen in cases than controls.

We highlight in our manuscript that both, *CFHR5* p.Gly278Ser and *CFHR5*_{fs} have arisen on haplotypes that also show additional variation in the *CFH/CFHR5* region. For instance, in Finns *CFHR5* p.Gly278Ser frequently ($r^2=0.85$) co-occurs with the missense variant p.Cys72Tyr (rs79351096; AF ~3%) in *CFHR2*, as well as three non-coding variants with previously assumed roles in driving AMD signals (Fritsche et al., 2016; Lorés-Motta et al., 2021). At the sample size of our recall study we cannot fully exclude that carrying *CFHR2* p.Cys72Tyr individually contributes to reduced FHR-2 levels in *CFHR5* p.Gly278Ser carriers. In fact, it is tempting to speculate that the co-occurrence of this missense variant explains why FHR-2 levels are even further reduced in *CFHR5* p.Gly278Ser carriers than in *CFHR5*_{fs} carriers, or carriers of the independently protective *CFH* alleles who do not carry coding *CFHR2* variants (**Supplementary Table 14**). Like *CFHR5* p.Gly278Ser, also *CFHR5*_{fs} is in high LD with ($r^2=0.91$) with five other regional variants. Notably, however, we demonstrate that 1.) none of the other known variants on the *FG*_{AMD3} and *FG*_{AMD4} haplotypes affects coding regions of nearby genes; 2.) based on haplotype-resolved sequencing in the 1000 Genomes Project there are no concealed coding or larger structural variants in the region that we could have overlooked; and 3.) deletion of *CFHR1*, *CFHR3* or *CFHR4* do not contribute substantially to the strong association between the *CFH* region and AMD in Finns. Thus, while our study cannot entirely exclude regulatory effects from low-impact non-coding variants, our results strongly propose variants in *CFH* and *CFHR5* genes as the most impactful contributors to AMD risk at the *CFH/CFHR5* locus. We have already highlighted in our original discussion that larger studies will be needed to clarify eventual additional contributions outside of *CFH* and *CFHR5* genes and now generally refer to the observed functional effects being caused by *CFHR5* loss-of-function *haplotypes* rather than the distinct *CFHR5* coding variants in isolation, even if these are most probably the strongest, if not sole contributors to the association signals.

Table 1 reflects the demographics of the recall study cohort, in which we intentionally enriched the number of heterozygous *CFHR5*_{fs} carriers with AMD by design. We assume that instead of Table 1 the reviewer refers to the table in **Figure 1b** which we have updated according to the reviewer's suggestions (see also our response to this reviewer's comment 23).

11. Lines 190-193: It is widely accepted that coding variants are of primary interest for causal variants, although they need to be validated by additional functional studies. Of course, a non-coding variant can also be an effective variant. Specifically, a non-coding variant may have regulatory effects on gene expression of any of the *CFHR1-5* genes. Such a situation was not addressed in this study. Thus, the statement "Because these are common deletion variants, this provides little support for the hypothesis that genetic variants impacting *CFHR1*, *CFHR3* and *CFHR4* contribute substantially to the strong association between the *CFH* region and AMD -

which our results suggest are primarily explained by variants in CFH and CFHR5 genes” seems too strong and should be rephrased.

This sentence has been rephrased as: *“Because these are common deletion variants, this provides little support for the hypothesis that variants impacting CFHR1, CFHR3 and CFHR4 coding regions contribute substantially to the strong association between the CFH locus and AMD, while conversely our results support a prominent role of high-impact variants in CFH and CFHR5 genes”*

12. Line 236: The term “pleiotropy” is used to refer to the expression of different phenotypic traits caused by a single gene. This does not appear to be the correct terminus technicus in the given context.

“Pleiotropy” has been substituted by “breadth of haplotypes”, which is indeed the better term in this context.

13. Lines 248-250: The statement “Taken together, our base-level interrogation of the CFH region excludes that the reduced risk for AMD in CFHR5 frameshift and missense variant carriers is conveyed by local SVs or concealed single base pair changes other than those impacting CFHR5” is incorrect as it ignores that the CFHR5 variants were detected as part of an association signal which per se is no proof of causal protective variants in CFHR5.

This sentence has been rephrased as: *“Taken together, our base-level interrogation of the CFH region finds no evidence to suggest that the association with reduced risk for AMD in CFHR5 frameshift and missense variant carriers is conveyed by local SVs or concealed single base pair changes other than those within CFHR5 or its associated haplotypes.”*

14. Lines 306-308: Again, the statement “This demonstrates that the protective effects of CFHR5fs on serum protein levels are independent of and add to that from other protective variation at the CFH locus” is an overinterpretation of the data. No independence was shown in the results. Notably, Supplementary Tables 12 and 13 indicate rather the opposite.

This sentence has been rephrased as: *“This demonstrates that CFHR5_{fs} conveys additive effects on serum protein levels that go beyond the effects from other protective variation on its respective CFH haplotype.”*

15. Lines 329-366 and Extended Data Figure 7: The results described in this paragraph are partially not in line with the data as given in the figures provided. For example, the postulated “...intermediate effects between homozygotes and non-carriers” are not visible in Extended Data Figure 7. In fact, the values of heterozygotes look similar to non-carriers. Also, in Figure 5c there is no significant difference between individuals carrying one CFHR5fs allele compared to individuals carrying no CFHR5fs allele (as described in the text in lines 347-352).

This sentence has been specified as: *“Heterozygotes significantly differentiated from non-carriers for both CP ($p=0.0071$) and AP ($p=0.0007$) only in AMD cases, but not in unaffected controls (Extended Data Figure 8), and in the overall cohort only showed a tendency towards intermediate effects between homozygotes and non-carriers (Figure 5b).”* For description of Figure 5C, a reference to heterozygotes has been omitted.

16. Lines 357-365: Although the authors state that “...results from these functional analyses demonstrate that genetic loss of CFHR5 function is independently associated with increased activation capacity of the complement system to confer protection from AMD”, independence was not shown in the performed functional analyses.

This sentence has been rephrased as: *“...results from these functional analyses demonstrate that the CFHR5_{fs} haplotype is independently associated with increased activation capacity of the complement system to confer protection from AMD”*

17. Lines 195-250: The frequency of CFHR5 variants is compared between Finns and non-Finnish individuals, however the latter group includes several related individuals. This results in biased allele frequencies between the two groups (see also Supplementary Table 4). In this context, it is unclear whether individuals included in investigating CFHR serum levels are related or unrelated.

This section discusses results from the 1000 Genomes Project (1KG) cohort. While the overall 1KG cohort indeed includes a substantial number of related individuals, all Finnish 1KG participants we identified as carrying a CFHR5 coding variant are annotated as “unrelated”. We have nevertheless removed any estimates of fold-enrichment in Finns versus non-Finns in 1KG from the revised manuscript to exclude the risk of biases.

Relatedness was not a pre-specified exclusion criterium for the recall study. However, since serum samples were chosen from a large historic collection of biospecimen stored at four different biobanks, each of which represents a vast geographic area within Finland, and since CFHR5_{fs} occurs with a frequency of 4% in Finns, we expect that the likelihood for an abundance of closely related individuals in the recall study cohort is small.

18. Lines 252-278 and Supplementary Table 9: The distribution of haplotypes for the two major independent signals should be summarized and mentioned in the main text. This should also clarify if the genotypes of the two major signals are evenly distributed between CFHR5_{fs} n/c, het and hom and the different AMD groups.

The distribution of all 13 CFH haplotypes represented in the FG recall cohort are now provided as a new **Supplementary Table 11** which we refer to in the main text. We would like to highlight that the recall cohort was designed primarily based on AMD case/control status (~1:1) and to enrich for carriers of CFHR5_{fs} (~30%). The distribution of haplotypes in this cohort is thus not representative for the haplotype distribution in the overall Finnish population. It includes some rarer haplotypes that in our population-scale studies we have not considered to determine the relative contribution of the CFHR5 haplotype to AMD risk on top of protective variation in CFH.

19. Lines 276-278 and Supplementary Table 10: In the text it is stated that no significant effect of FHR1 was observed for a p-value cutoff of 0.05. In Supplementary Table 10, however, the respective p-value is 0.03 and therefore below the threshold. The same is true for the AMD-Status. Applying the same p-value cutoff would result in significant results for FHR-2 and FHR-5. This would indicate that the AMD status has an effect on FHR-2 and FHR-5 serum levels which is important as the distribution of hetero- and homozygous carriers of the frameshift variant are not evenly distributed between AMD cases and controls.

We did not to emphasize results that were borderline without independent replication data. In our revised manuscript we now highlight that also FHR-1 levels change at nominal significance ($p=0.03$) in *CFHR5_{fs}* carriers in the recall cohort, but that we were unable to replicate this effect by SomaScan analyses in the wider FG cohort ($p=0.113$ and $p=0.495$, respectively, for both FHR-1 somamers available).

Since association analyses with AMD status are confounded by the strong enrichment for the *CFHR5_{fs}* haplotype in the recall study cohort, we have decided to remove the sentence that FH and FHR1-5 levels were not independently associated with AMD after controlling for *CFHR5_{fs}* carrier status to avoid any risk of overinterpretation. However, we have now added a paragraph to the **Discussion** that examines our results in the light of earlier studies reporting associations between elevated levels of certain FHRs with increased in AMD risk (e.g., Cipriani et al., 2021; Emilsson et al., 2022) and lower FHR levels with protection from AMD (e.g., Lorés-Motta et al., 2021).

20. Line 335-345 and Figure 5b: Samples sizes of the three different genotypes should be given.

Sample sizes have now been added to the Legend to **Figure 5b**.

21. Supplementary Table 12: Data indicate that the major independent signals in the CFH locus also influence serum levels. Unfortunately, the information in the Methods is insufficient to judge if adjustment for independent signals which occur in the respective individuals was performed in the analysis. Also, FHR-1 reveals a significant result for the *CFHR5* frameshift variant, while FHR-4 fails to reach significance. In fact, this is consistent with Supplementary Table 13, where an effect for FGAMD1 and FGAMD2 was observed for several FH and FHR serum levels.

Multi-variant analyses in Supplementary Table 12 (now **Supplementary Table 14**) were controlled for age, sex, AMD status and the four alleles under study, respectively. We are now providing this information in **Methods** and in the legend to the Supplementary Table. Indeed, both our results from the recall study (**Supplementary Table 14**) as well as proteomics analyses in the wider FinnGen (**Supplementary Table 15**) and UK Biobank populations (**Supplementary Table 16**) confirm that *CFH* variants, incl. rs1410996 and p.Tyr402His, impact blood levels not only of FH, but also several of the FHR proteins. Such effects have already been reported previously (e.g., Lorés-Motta et al., 2021; Emilsson et al., 2024). For example, Emilsson et al. found that rs10922109_A, which confers protection against AMD, was associated with lower levels of FHR-

4 (see also Reviewer #4, comment 2). Consistently, we observed this association also in FinnGen with both the Olink (n=1,732) and the SomaScan platforms (n=881). We now more thoroughly compare results from our analyses to these previous studies in our **Discussion**.

22. Reference 17 is highly relevant to the current study and should be critically included in the Discussion section.

Results from Lorés-Motta et al., 2021 are now discussed more extensively.

23. Supplementary Table 2: Confusing mix of data from Fritsche et al. 2016 and own data. This should be resolved and presented in a structured way to make the point clear to the reader.

Supplementary Table 2 (now **Supplementary Table 3**) and accompanying **Figure 1b** have been restructured and should now make our own relative to previous work clearer to the reader.

24. Supplementary Table 4: Abbreviations should be explained. Heterozygous alleles are given in the form 0/1 or 1/0. This should be harmonized.

Supplementary Table 4 has been updated accordingly.

25. Supplementary Table 6: unclear if columns indicated with AC refer to individuals/haplotypes. If so, why are several individuals/haplotypes included more than once? For example, AC=11 occurs four times and always with different variants.

Previous Supplementary Table 6 (now **Supplementary Table 7**) lists 254 different haplotypes in the *CFH* region that we identified through haplotype-resolved sequencing of the 1000 Genomes (1KG) cohort. These haplotypes include either our FG lead signals, structural variants with AF>1% and/or any of the eight lead SNPs that have been proposed previously as likely independent from published AMD GWAS (Fritsche et al., 2016; Lorés-Motta et al., 2021). “AC” was supposed to refer to “allele count”. To avoid confusion, we have now changed this term to “HC” (haplotype count) and explain the abbreviation.

26. Supplementary Table 7: Terms “FIN%FIN” and “FIN%ALL” should be clarified.

“FIN%FIN” designates the proportion of the respective haplotype among all Finnish haplotypes in the 1KG cohort; “FIN%ALL” designates the proportion of Finns among all carriers of the respective haplotype in the 1KG cohort. We have now clarified this in the table (now **Supplementary Table 8**).

27. Supplementary Table 8: Heading needs to be changed from Supplementary Table 10 to Supplementary Table 8.

Done. The respective table is now **Supplementary Table 9**.

28. Methods: It is unclear how UK Biobank data were analyzed and how the burden for coding variants was determined.

For clarification, the following sentence in **Methods** has been rephrased as: “*To identify plasma proteins associated with the burden of protein-coding variants in CFH and CFHR1-5 genes we queried results from Dhindsa et al 2023²⁹ who had linked whole exome sequencing data to protein levels obtained with the Olink panel from plasma of 49,736 individuals of European ethnicity as part of the UK Biobank Pharma Proteomics Project. The respective results from this study are provided as **Supplementary Table 16.***”

29. Line 562: Nomenclature for the HTRA1 variant 10:122461685:G:GTCCT should read 10:122461685:C:TCCT.

The nomenclature of this variant has been updated to 10:122461686:C:TCCT.

30. Figure 1: In Fig. 1a signal 1.5 is missing. Also, the missense variant Tyr402His should be located in an exon of CFH. Why are some genes highlighted in green?

Figure 1a has been updated to hg38 and now also includes signal 1.5. We have clarified in the Figure legends that the variants highlighted are the high-LD proxy variants used to impute the causal variants in FG, thus *CFH* p.Tyr402His is represented by the intronic rs570618 (FG_{AMD2}). All genes are now marked up in the same color.

31. The terms FGAMD3 and FGAMD4 are mixed in the text with the descriptions missense and frameshift variant, respectively. Maybe it would help the reader to refer to a single terminology.

We are now only using the “FG_{AMD}1-4” nomenclature when describing the finemapping results. In all other sections of the manuscript “*CFHR5* p.Gly278Ser” (or “*CFHR5* missense”) and *CFHR5*_{fs} haplotypes are now utilized.

32. Figure 2a: It is unclear why Fig. 2a annotates a SNP twice with “*CFHR5*p.Glu163insA”.

Thank you for noticing this typo. Corrected.

33. Table 1: AMD cases add up to 200 (not 199 as indicated in the last column).

Table 1 has now been corrected.

34. Legend to Figure 5: It is unclear why a student’s t-test was performed and for which analysis? The data show no two group comparisons.

Legends to **Figure 5** and **Extended Data Figure 8** have been corrected.

Reviewer #2 (Remarks to the Author):

We thank Reviewer #2 for co-reviewing our manuscript and supporting NCOMM's initiative to formally recognize the contributions of early career researchers during the review process.

Reviewer #3 (Remarks to the Author):

This study describes a GWAS for AMD in FinnGen, a unique founder population that enables identification of rare variants and can pinpoint causal genes. Of particular interest are a rare coding variant in the HTRA1 gene suggesting a causal role of HTRA1 in AMD, in addition to rare frameshift variant in CFHR5 at the CFH locus. Circulating protein levels of 6000 proteins, including FH and FHR-1-5, were measured using the Somascan platform. Notably, not only CFHR5, but also CFHR2 and CFHR4 were decreased in carriers of the CFHR5 frameshift variant while complement activity levels were increased.

1. The CFHR5 frameshift variant was found to be protective for AMD, and is associated with reduced FHR-5 levels using the Somascan platform. Surprisingly complement activity (AP and CP) levels are increased in carriers of the CFHR5 frameshift variants (Figure 5). Increased complement activation levels have consistently been described to lead to increased risk for AMD, therefore, this finding is highly unexpected. The conclusion "Our findings ... propose therapeutic downregulation of FHR-5 as a promising strategy for prevention or treatment of AMD" is therefore incorrect. A therapeutic strategy for downregulating FHR-5 would only be warranted if genetic loss of CFHR5 would be associated with decreased risk of AMD and decreased complement activity levels. This questions the validity of the conclusions drawn from the AP and CP activity assays used in this study; further investigation using other commonly used and more robust assays to assess complement activity is warranted (e.g. C3b degradation and/or C3d/C3 ratio), in addition to in vitro studies to assess the isolated effect of CFHR5 variants using recombinantly expressed CFHR5 protein on complement activity. Another approach to assess the validity of the AP and CP activity assays is to evaluate the effect of the CFH Y402H risk variant on complement activity, as these levels are expected to be increased by this risk allele.

We thank the reviewer for their interest in our findings and thorough reflections on our manuscript. We would like to clarify that throughout our manuscript we were *not* aiming to report complement *activation* in *CFHR5_{fs}* carriers as *increased*, but rather that serum samples of *CFHR5_{fs}* carriers show an *increased activity* (= *activation capacity*) of the CP and AP. If complement has become activated in a sample, the *activation capacity* has *decreased*, and the levels of activation products are increased. In our manuscript we used the standardized WIESLAB test (WIESLAB Complement System Screen COML 300; Euro-Diagnostica, Malmö, Sweden), an assay that is commonly applied to assess the function of the three branches of the complement system both, in research and clinical settings. This assay allows the measurement of *total complement activation capacity in the samples* (If one would compare the complement system to driving a car

this would be analogous to “*how much gas is in the tank*”). Our reasoning is that in the absence of an activator, here *CFHR5*, complement activation capacity is preserved and thereby can be measured at a higher level (“*less gas has been used*”). Complement activation means a situation where complement components become activated and are thus *consumed*. This occurs, if the balance between activation and regulation shifts to the direction of activation, for instance in the presence of an activator (“*the gas pedal is being pressed*”). FHR-5 is such an activator: it counteracts the inhibitory function of factor H, so in its presence the balance shifts towards *activation* and complement consumption. This leads to the emergence of complement activation fragments, like iC3b, Bb and the terminal complement complex which are harmful to the retina. Consequently, therapeutic downregulation of FHR-5 is expected to reduce the emergence of such fragments and result in beneficial effects on the retina, as we see in *CFHR5_{fs}* haplotype carriers in our study. We have now clarified more precisely what exactly our assay measures, have updated **Figure 5** and **Extended Data Figure 8** Figure legends and y-axis labels, and apologize for any confusion.

With regards to carriers of the *CFH* Y402H *risk allele*, they would be expected to have lower complement activity (= *activation capacity*) because of complement activation and consumption, whereas carriers of the respective protective allele should show the inverse. If our reasoning above is correct, the latter should be reflected in a higher preserved complement activation capacity. Notably, this is exactly what we observe with our assay when we break down the recall study participants into distinct haplogroups according to whether they carry AMD-protective alleles at rs1410996 and/or p.Tyr402His (n-n-0-0) versus not (0-0-0-0). While our recall study was powered to test the functional effects of the *CFHR5_{fs}* haplotype rather than that of the more common, but much less impactful AMD-associated variants in *CFH*, we still see that *CFH* protective variant carriers show statistically significant higher levels in AP activation capacity ($p=0.018$). In AMD cases, both, CP ($p=0.046$) and AP ($p=0.034$), but not LP ($p=0.26$) activation capacities are increased. These findings validate the reviewer’s hypothesis that *CFH* p.Tyr402His would show an effect in our assay and further support that indeed therapeutic *downregulation* of FHR-5 would be the most promising strategy to prevent or treat AMD. We are now describing these new findings in **Methods** and have added a new **Extended Data Figure 9**.

2. Notably, the *CFHR5* frameshift variant was found to be protective for AMD, while a *CFHR5* missense variant (FG-AMD3) was found to be associated with increased AMD risk (Figure 4). This further complicates the conclusion that loss of *CFHR5* is associated with decreased risk of AMD; the *CFHR5* frameshift variant supports this conclusion but the *CFHR5* missense variant does not.

We would like to clarify that our results show the minor allele of *CFHR5* p.Gly278Ser (rs139017763_A) to be *protective* for AMD. This allele can be found in the Finnish population with an allele frequency of ~2%. It can be imputed with high confidence ($r^2=0.98$) from the independent FinnGen lead variant rs537634973 (FG_{AMD3}) where the minor T-allele is associated with a strong protection from AMD ($p=6.2 \times 10^{-25}$ after conditioning on FG_{AMD1+2}). Notably, p.Gly278Ser is in high linkage disequilibrium with an independent protective association described in Fritsche et al., 2016 (**Supplementary Table 3**). With this, both the missense and the frameshift variant convey

protection from AMD. We would like highlight though that since the missense and frameshift variants do not co-occur on the same haplotype in Finns, **Figure 4** may have given the erroneous impression that they act in inverse directions, which is not the case. We apologize for any confusion.

3. The authors refrained to evaluate the effect of the rs800292 (V62I) variant in the CFH gene leading to a valine to isoleucine change at amino acid position 62 of the CFH protein, which has been postulated to be protective for AMD by prominent researchers in the AMD field. An assessment of common risk and protective variants at the CFH locus, in particular the rs800292 coding (V62I) variant in the CFH gene, on CFH and CFHR protein levels may help clarify an ongoing debate related to the studies published by Zouache et al (Nat Commun 2024 Jan 10;15(1):443) and Cipriani et al (Nat Commun. 2020 Feb 7;11(1):778.). How does the V62I variant relate to the evaluated CFH haplotypes; is this variant independently associated with AMD, and what is the effect of this variant on CFHR protein levels and complement activity?

The *CFH* missense variant Val62Ile (rs800292; 1:196673103-G-A) has a prevalence of nearly 30% in the Finnish population. We have reported previously, that carrying the protective Ile62 allele is associated with higher levels of AP, but not CP activity (=activation capacity) (*Salminen et al., Circulation: Cardiovascular Genetics* 2017), but did not investigate whether this effect was independent of co-associated variation at the *CFH* locus. In FG DF12 univariate analysis the variant is associated with a strong protection from AMD (uncorrected $p=2.4e^{-282}$) and AMD co-associated phenotypes (<https://r12.finngen.fi/variant/1:196673103-G-A>). However, while not in particular high LD with the independent *CFH* lead variants, the association signals from this variant are not conditionally independent after correcting for the two highest-significant *CFH* signals (FG_{AMD1} and FG_{AMD2}). At least in Finns, our multistep conditional analyses thus show no indication for the *CFH* V62I variant to be independently associated with AMD.

Reviewer #4 (Remarks to the Author):

Reeve et al. demonstrate through fine mapping and conditional analyses that four independent haplotypes in the CFH genomic region on chromosome 1 confer protection against age-related macular degeneration (AMD) in Finns, as shown in the large-scale FinnGen genetic cohort. Two of these haplotypes, which reflect deleterious variants within the CFHR5 gene, are enriched in the Finnish population compared to another European cohort (UK Biobank data). These haplotypes are associated with lower levels of CFHR5 (FHR-5), as well as CFHR2 (FHR-2) and CFHR4 (FHR-4), and correlate with increased activity of relevant complement system pathways. The authors had previously reported a frameshift variant within CFHR5 (Glu163insAA, rs565457964) that provides protection against AMD, meaning the protective effect conferred by the new haplotypes is not entirely novel. Nevertheless, the identification of these four independent genetic signals within the complex AMD linked region on chromosome 1 represents new information for the field.

Main Point of Criticism:

1. The main issue with this manuscript is its lack of novelty. As mentioned, the authors previously published the findings of a frameshift allele in *CFHR5* that was protective for AMD. In addition, other researchers published work in *Nature Communications* (PMID: 35697682), that documented several aspects of the relationship between the protein *CFHR5* (aka *FHR-5*) and AMD. This included its involvement in late-stage AMD, its potential to predict advanced AMD, and its causal relationship with advanced AMD, as demonstrated through two-sample Mendelian randomization analysis. This work was unfortunately not cited in this manuscript. These two reports have already established a link between *CFHR5* and AMD, and as such there is little novelty to Reeve et al.'s findings.

We thank the reviewer for directing our attention to the very informative publication by Emilsson et al., 2022 which is now cited and discussed in our manuscript. We apologize for the earlier omission to reference this important study. In their study, the authors analyzed how levels of 4,782 serum proteins as measured by the SomaScan platform from 5,457 participants of the Icelandic AGES-RS cohort relate to different stages of AMD and mediate the genetics of disease. They identify 28 proteins as significantly associated with AMD. Among these, elevated levels of *FHR-1* and *FHR-5* were found as predictive for progression to later stages of disease. MR-Egger analyses support that this effect is mediated through cis-regulatory genetic variants at the *CFH* locus. With their use of cis-acting instrumental variables from GWAS for MR-Egger, Emilsson et al. support previous findings by Cipriani et al., 2021 who had used a Wald-ratio based single variant model for MR to demonstrate a link between regional GWAS variants, elevated *FHR-5* levels and higher risk for advanced AMD. They also support findings by Lorés-Motta et al., 2021 who showed that certain low-frequency genetic variants in *CFHR5* that are associated with reduced *FHR-5* levels are also associated with protection from AMD. They further add an additional dimension to our own earlier findings that carriers of the rare *CFHR5* p.Glu163insAA (*CFHR5_{fs}*) variant show a strong protection from AMD (Sun et al., 2022).

We agree with the reviewer that these previous studies already hint at *CFHR5* and its protein product as putatively contributing to the strong AMD signal at the *CFH* locus and, when taking their findings together, that *CFHR5* could be a potential AMD-protective allelic series gene. However, none of these earlier works thoroughly assessed the effects driven by *CFHR5* variants in relation to the multifold other variation at the *CFH* locus, most probably, as Emilsson et al. state, since “*the loc[us] containing ... CFHR5 [is] saturated with multiple independent variants for both AMD and proteins, colocalization analysis becomes difficult*”.

In our current study, we carefully dissect the *CFH* locus through multiple rounds of finemapping and conditional analyses in Finns that due to a reduced complexity of the regional haplotype structure and enrichment for *CFHR5* coding variants *CFHR5_{fs}* and *CFHR5* p.Gly278Ser are particularly advantageous for deconvoluting association signals at this complex GWAS locus. Our results firmly establish that carriers of haplotypes with *CFHR5* loss-of-function alleles show an additive protection from AMD on top of that conveyed by protective alleles in *CFH*, particularly rs1410996_A and p.Tyr402His (rs570618) which explain the majority of association signal at the *CFH* locus. We further demonstrate through dedicated follow-up analyses that carriers of the *CFHR5_{fs}* haplotype show reduced *FHR-5* protein levels and an enhanced activation capacity of

the classic and alternative complement pathways. Moreover, in our revised manuscript we now also report that carriers of the *CFHR5*_{fs} haplotype show protective changes in the photoreceptor layer in their retinas as measured from optical coherence tomography (OCT) scans in UK Biobank, an emerging surrogate endpoint for early AMD. The pronounced effects we observe across multiple domains in *CFHR5* loss-of-function haplotype carriers cannot be explained by any other cryptic variation of protein-coding sequences at the *CFH* locus, making *CFHR5* highly probable as an independent modulator of genetic risk and a promising drug target for AMD.

Our revised **Discussion** now better describes the novelty of our findings in relation to the literature.

2. The main claimed point of the manuscript is the clear identification of *CFHR5* as the culprit in AMD. Yet the protective *CFHR5* haplotypes were not only associated with lower *CFHR5* (FHR-5), but *CFHR2* (FHR-2) and *CFHR4* (FHR-4) as well. In other words, this reflects molecular pleiotropy that brings into question the authors main claim. The authors seem to be somewhat selective when discussing this point. While they frequently suggest that targeting *CFHR5* (FHR-5) could be a novel therapeutic strategy for AMD, their results also support a strategy of lowering *CFHR2* (FHR-2) and *CFHR4* (FHR-4) levels as well. In fact, the previously mentioned missing citation (Nature Communications, PMID: 35697682) found that the *CFH* variant rs10922109_A allele, which confers protection against AMD, was associated with lower levels of *CFHR4* (FHR-4) protein. This aspect of their work should have been acknowledged, as its omission undermines the impact of their findings.

Indeed, our results from both, recall study participants as well as a wider FinnGen cohort show that carriers of *CFHR5* protective haplotypes not only show reduced FHR-5, but also FHR-2 and FHR-4 serum levels. These findings are independently supported by gene-based burden analyses in UK Biobank. Conversely, serum levels of FH and FHR-3 do not show apparent changes at our sample sizes and with the proteomics platforms utilized to measure FH and FHRs.

We tested whether the pronounced reduction in FHR-2 and FHR-4 levels might be explained by variants affecting coding regions in these genes through analysis of haplotype-resolved long-read sequencing data of the *CFH* region from the 1000 Genomes cohort. However, we did not find evidence for previously unknown rare or cryptic variants in these genes to co-occur with *CFHR5*_{fs} status. Also, prior evidence suggests that at least *CFHR4* is not a substantial contributor to AMD risk or progression at the *CFH* locus (e.g., Zouache et al., 2024). To avoid overinterpretation of our results, we now discuss in the manuscript that we cannot fully exclude a contribution of co-occurring non-coding variants to an altered expression of these genes in relevant tissues. However, our most likely explanation for lower FHR-2 and FHR-4 levels in the absence of FHR-5 is an increased turnover at the protein level because of the lack of a binding partner to form a stabilizing complex. Future studies will be needed to experimentally test this hypothesis and how specific for FHR-5 potential drug candidates may need to be.

In our analyses, carriers of rs10922109_A, the *CFH* intronic variant most strongly associated with protection from AMD also in our study (FG_{AMD}1), show increased FH, reduced FHR-2 and

nominally ($p=0.05$) reduced FHR-4 levels in our recall cohort, as well as increased FH and reduced FHR-1, FHR-2, FHR-4 and FHR-5 levels in the wider FG cohort. We now acknowledge the earlier finding of a relationship between rs10922109 and FHR-4 and also discuss, that differences between studies assessing FHR serum levels are likely explained not only by differing sample sizes, but also variation in haplotype composition of the respective study cohort.

Additional Points of Criticism:

3. FinnGen, a public resource, is one of the many prominent modern genetic cohorts utilizing large datasets to enhance statistical power in detecting rare variants associated with disease. An important limiting factor that cannot be solely addressed by increasing sample size, however, is the quality and depth of phenotype annotation and measurements within the cohort. Comprehensive phenotypic data provides more precise insights, enabling researchers to distinguish between meaningful genetic signals and potential confounders. It is unclear from the manuscript to what extent the AMD population in this study is characterized in terms of different AMD types, stages, and progression, including the transition from early to late stages, such as geographic atrophy or the more severe neovascular form of AMD. Associating the different haplotypes with these measures would be of significant interest to the field. A fuller discussion of this point would have been appreciated.

For the present study, we have deliberately chosen a wide definition of AMD, specifically FinnGen endpoint H7_AMD, to maximize the power for gene discovery and downstream validation analyses. H7_AMD (https://risteys.finregistry.fi/endpoints/H7_AMD) includes both, wet and dry AMD which have both been analyzed individually in FinnGen (see https://r12.finngen.fi/pheno/DRY_AMD and https://r12.finngen.fi/pheno/WET_AMD). For our recall study, we selected stored serum samples from FG participants with at least one diagnostic entry for dry AMD (H35.30, 2625A), but no entry for wet AMD or ICD codes for hereditary conditions affecting the eye (e.g., Stargardt disease, hereditary retinal dystrophy), macular hole, macular pucker, diabetic retinopathy, glaucoma, vitreous hemorrhage, separation of retinal layers, onset of blindness at <30 years of age, or injections into the eye. Controls were required to have no ICD entries for AMD, be 65 years of age or older at time of sampling and were matched to cases based on age and sex. To download a full list of clinical endpoints utilized for FG DF12, as well as a more extended list of at least 258 endpoints reflecting diseases of the eye and adnexa (H7_) that are in principle available to the FG project, but not part of FG core analyses, see here: <https://www.finngen.fi/en/researchers/clinical-endpoints>. Partners within the FG project can customize these endpoints according to epidemiological parameters, time since diagnosis, or co-medications, among others (see e.g., <https://risteys.finngen.fi>) in a dedicated research environment. Thus, while often not entirely living up to the quality and depth of data ascertained in a disease-specific manner (e.g., as part of interventional clinical trials) there is a tremendous potential to utilize the FG cohort for more refined analyses in follow-up studies. We are referring to our revised **Discussion** and our earlier publications on FG for a description of this potential in more details.

4. Furthermore, the relationship between protein levels and disease risk is not always symmetric: while an increase in the level of a protein that raises disease risk might imply that inhibiting or

reducing its activity could lower the risk, this is not always the case due to factors such as compensatory mechanisms and natural feedback loops. Therefore, demonstrating that elevated CFHR5 levels increase the risk of AMD is just as novel as showing that reduced levels offer protection, and may indicate that in this instance, symmetry might apply. A better discussion of this in the context of all prior published work would have been a nice addition too.

In our revised **Discussion**, we are now featuring better that elevated levels of FHR-5 have been found as predictive for and possibly causally involved in the pathogenesis of AMD as has been reported previously (e.g., Cipriani et al., 2021; Emilsson et al., 2022). We agree that the apparent symmetry of elevated levels of FHR-5 being associated with increased risk for AMD while reduced levels of FHR-5 are associated with a lower risk for AMD further enhance the attractiveness for dedicated follow-up on *CFHR5* and its suitability as a drug target.

5. Page 13, lines 451-452. “With the emergence of both sets of variants on the same haplotype, evolution may have found a mechanism to re-adjust the balance between FH and FHR-5 in a most favorable way to prevent disease”. How does this sentence make any sense, given that negative (purifying) or positive selection forces typically act to preserve or enhance reproductive fitness, yet AMD (assuming this is the disease referred to) primarily affects older adults, who are in the post-reproductive age?

We have now rephrased this sentence as: “*With the emergence of both sets of variants on the same haplotype, evolution may have found a mechanism to re-adjust the balance between FH and FHR-5 in a most favorable way to optimize the complement gene family for its primary function in pathogen recognition and host immune defense, which could secondarily benefit the emergence of late-onset diseases such as AMD (Cantsilieris et al., 2018).*”

6. A related, though potentially less relevant question considering the rarity of the protective CFHR5 haplotypes: given that these haplotypes are enriched in Finns, likely due to population bottleneck and/or genetic drift, are there any epidemiological studies that show whether the prevalence or incidence of AMD differs between Finns and other European populations?

Based on FinRegistry data from health records of ~5 million Finns (see: https://risteys.finngen.fi/endpoints/H7_AMD), the unadjusted period prevalence of broadly defined AMD is 1.64% in the overall Finnish population and 2.48% among the (on average older) FinnGen population. This prevalence rises steeply with older age, and among 90-year-olds in Finland, 15.1% of women and 10.0% of men have received a registry-based diagnosis of AMD. Consistent with epidemiological information from other countries, the median age at diagnosis in Finns is 79.03 years and 76.26 years in FinnGen. With this, the epidemiology of broadly defined AMD in Finns does not seem to differ substantially from that of progressed AMD in other European populations with comparable age distributions. See also our response to Reviewer #1, comment 6.

7. While revising the manuscript by Reeve et al., the largest multi-ancestry GWAS meta-analysis of genetic risk factors for AMD was published (Nature Genetics, PMID: 39623103), identifying

many new loci associated with AMD risk. Reeve et al. report that 23 risk loci (including 6 new loci) for AMD have been identified through the FinnGen study. Considering this, the authors should alter their manuscript to clarify the number of new loci identified in FinnGen in comparison to this more comprehensive genetic study of AMD risk.

For the revised manuscript, we updated our comparisons based on the recently published multi-ancestry meta-GWAS by Gorman et al. (61,248 AMD cases and 364,472 controls). As we highlight in **Results**, 21 of the 23 loci significant in FG fall into regions that have been reported previously as associated with AMD or its subforms by either Fritsche et al. or Gorman et al., while two appear to be novel (rs759283, rs74026308). Of the 62 loci identified as associated with AMD in that study, 19 loci met genome-wide significance criteria in FG DF12, 31 replicated at nominal significance ($p < 0.05$) and 11 did not replicate (although 10 of the 11 appear to be in the same direction as in Gorman et al., with one locus not being included in the FG analysis set). We have annotated the FG association results with information from Fritsche et al., 2016 and Gorman et al., 2024 in a revised **Supplementary Table 1** and added a new **Supplementary Table 2** that provides FG association results for all AMD loci described by Gorman et al.

8. Why do the authors use different annotations for the gene (CFHR5) and its protein product (FHR-5)? The protein annotation seems to be based on historical usage and may no longer be up to date.

Since our study assesses both, gene and protein level, as well as the interplay between the two, we believe that keeping genetic and protein nomenclature separate will improve clarity of our manuscript.

Point-by-point response to reviewers:

Reviewer #1 (Remarks to the Author):

The authors have undertaken a comprehensive revision of the manuscript, thoroughly addressing all points raised by Reviewer 1. These revisions have significantly improved the readability and conclusiveness of the study. Notably, the authors have incorporated extended data tables and figures and have substantially expanded the Discussion section, including additional references. Furthermore, their refined interpretation of the data is acknowledged and appreciated. These improvements enhance the overall rigor and comprehensiveness of the study. Two points remain that the authors may consider addressing:

1. AMD Prevalence in Finns (Reviewer 1, Former Point 6)

It would be beneficial to include information on AMD prevalence in Finns within the manuscript itself rather than solely in the rebuttal letter. Providing this context would enhance the interpretability of the findings for readers.

We are glad that our revised manuscript has addressed the reviewer's concerns and again would like to thank them for their help in improving important details in our manuscript. We have now added the information on AMD epidemiology from the rebuttal letter to the Methods section of our manuscript where it now reads as such:

Based on FinRegistry data from health records of ~5 million Finns (see: https://risteys.finnngen.fi/endpoints/H7_AMD), the unadjusted period prevalence of broadly defined AMD is 1.64% in the overall Finnish population and 2.48% among the (on average older) FinnGen population. Consistent with epidemiological information from other countries, this prevalence rises steeply with older age, and among 90-year-olds in Finland, 15.1% of women and 10.0% of men have received a registry-based diagnosis of AMD. The median age at diagnosis in Finns is 79.03 years and 76.26 years in FinnGen. Thus, the prevalence of broadly-defined AMD in Finland does not seem to differ substantially from the prevalence of progressed AMD in other European populations with comparable age distributions, although we recognize that direct comparisons across national cohorts can be biased (Colijn et al., 2017).

2. Allele Frequency Comparison and Related Individuals (Reviewer 1, Former Point 17)

The removal of fold-enrichment estimates for Finns versus non-Finns is noted and appreciated. However, a key concern remains. While our initial critique may have been misinterpreted, comparing allele frequencies between a cohort that includes related individuals and one that does not may introduce bias, particularly given the limited sample size. A more rigorous approach would involve excluding related individuals from the 1000 Genomes (1KG) cohort by filtering based on family ID and recalculating allele frequencies. While this adjustment would reduce the sample size, it would yield more comparable and less biased allele frequency estimates.

We apologize for any possible misinterpretation of the reviewer's initial critique. As proposed, we have now re-calculated frequencies of unrelated *CFHR5* variant carriers among just the unrelated participants in the 1KG cohort. This sub-cohort comprises 934 participants (1,868 chromosomes) versus 3,202 participants (6,404 chromosomes) in the full cohort. It includes 294 Europeans of which 99 are Finns. All 19 *CFHR5* coding variant carriers in this cohort are of European ancestry. Of the twelve Finnish individuals in this group, six carry *CFHR5* p.Gly278Ser (*CFHR5*_{mis}; out of ten total) and six p.Glu163insAA (*CFHR5*_{fs}) (*CFHR5*_{fs}, incl. one homozygote; out of eight total). The only unrelated individual carrying the p.Glu163insA frameshift variant was non-Finnish. We have re-calculated *CFHR5* coding variant frequencies based on these numbers for the relevant

subcohorts among unrelated 1KG participants and provide the updated frequencies now as a new **Supplementary Table 6**. Our initial observations that $CFHR5_{fs}$ and $CFHR5_{mis}$ are found in 3-4% of Finnish chromosomes, but are much rarer in other 1KG ethnicities, remain unchanged.

Supplementary Table 6: Frequency of CFHR5 coding variants in unrelated 1KG participants (n=934 individuals)

CFHR5 variant	carrier frequency in unrelated 1KG participants					allele frequency in unrelated 1KG participants				
	overall (n=934)	EUR (n=294)	FIN (n=99)	all-FIN (n=835)	NFE (n=195)	overall (n=1,868)	EUR (n=588)	FIN (n=198)	all-FIN (n=1,670)	NFE (n=390)
p.Gly278Ser (n)	0.0107 (10)	0.034 (10)	0.0606 (6)	0.0048 (4)	0.0205 (4)	0.0054 (10)	0.017 (10)	0.0303 (6)	0.0024 (4)	0.0103 (4)
p.Glu163insAA (n)	0.0086 (8)	0.0272 (8)	0.0606 (6)	0.0024 (2)	0.0103 (2)	0.0048 (9)	0.0153 (9)	0.0354 (7)	0.0012 (2)	0.0051 (2)
p.Glu163insA (n)	0.0011 (1)	0.0034 (1)	0 (0)	0.0012 (1)	0.0051 (1)	0.0005 (1)	0.0017 (1)	0 (0)	0.0006 (1)	0.0026 (1)

We have also updated the respective section in Results (changes underlined):

*Consistent with our previous findings²³, $CFHR5$ p.Glu163insAA was substantially more prevalent in unrelated Finns (7/198; AF=3.54%) relative to all other unrelated 1KG participants (2/1,670; AF=0.12%) (**Supplementary Table 6**), with the only non-Finnish carriers being two Tuscans from Italy and one African American individual. Also p.Gly278Ser is enriched in Finns (6/198; AF=3.03%) relative to the unrelated 1KG cohort (4/1,670; AF=0.24%), while only one of the rs191281603-G and none of the p.Glu163insA carriers were Finnish.*

Reviewer #2 (Remarks to the Author):

We thank Reviewer #2 for their contributions to co-review our manuscript.

Reviewer #3 (Remarks to the Author):

Thank you for addressing the concerns of the reviewers, and for clarifying several areas that were unclear.

Although the manuscript has improved with the edits, I do not see much novelty of the findings described in the manuscript. As also pointed out by reviewer 4, the authors describe $CFHR5$ variants that have been previously described to have a protective effect on AMD, and that have previously been shown to reduce FHR-5 levels (Emilsson et al 2022, Nat Commun 13:2401; Lores-Motta et al 2021, AJHG 108:1367).

The authors have also not attempted to clarify the ongoing debate related to the studies published by Zouache et al (Nat Commun 2024 Jan 10;15(1):443) and Cipriani et al (Nat Commun. 2020 Feb 7;11(1):778.), while this could have been a strength of this manuscript.

As we have highlighted in our original response to Reviewer #4, who now considers our changes as addressing their comments comprehensively and has no further remarks, no previous study has provided a similar level of evidence for $CFHR5$ as a second key driver of the massive AMD association signal within the CFH region. Likewise, no other study has assessed the effects driven by $CFHR5$ variants in relation to the multifold other variation at the CFH locus at comparable sample sizes and with similar rigor.

In our manuscript, we carefully dissect the CFH locus through multiple rounds of finemapping and conditional analyses in Finns that due to a reduced complexity of the regional haplotype structure and enrichment for $CFHR5$ coding variants $CFHR5_{fs}$ and $CFHR5$ p.Gly278Ser are particularly advantageous for deconvoluting association signals at this complex GWAS locus. Our results

firmly establish that carriers of haplotypes with *CFHR5* loss-of-function alleles show an additive protection from AMD on top of that conveyed by protective alleles in *CFH*, particularly rs1410996_A and p.Tyr402His (rs570618) which explain the majority of association signal at the *CFH* locus. We further demonstrate through dedicated follow-up analyses that carriers of the *CFHR5*_{fs} haplotype show reduced FHR-5 protein levels and an enhanced activation capacity of the classic and alternative complement pathways. We also report that carriers of the *CFHR5*_{fs} haplotype show protective changes in the photoreceptor layer in their retinas as measured from optical coherence tomography (OCT) scans in UK Biobank, an emerging surrogate endpoint for early AMD. The pronounced effects we observe across multiple domains in *CFHR5* loss-of-function haplotype carriers cannot be explained by any other cryptic variation of protein-coding sequences at the *CFH* locus, making *CFHR5* highly probable as an independent modulator of genetic risk and a promising drug target for AMD.

As we are detailing in our previous response to this reviewer, the association signals from the *CFH* missense variant Val62Ile (rs800292; 1:196673103-G-A) are not conditionally independent after correcting for the two highest-significant *CFH* signals (FG_{AMD1} and FG_{AMD2}). Thus, our multistep conditional analyses in Finns show no indication for the *CFH* V62I variant to be independently associated with AMD.

Reviewer #4 (Remarks to the Author):

I believe the authors have addressed my comments comprehensively, and I have no further remarks.

We thank the reviewer for their help in improving our manuscript.